# TrigReason: Trigger-Based Collaboration between Small and Large Reasoning Models

## Abstract

Large Reasoning Models (LRMs) achieve strong performance on complex tasks through extended chains of thought but suffer from high inference latency due to autoregressive reasoning. Recent work explores using Small Reasoning Models (SRMs) to accelerate LRM inference, yet existing frameworks such as SpecReason adopt a polling-based design that repeatedly invokes the LRM for verification at every step. This approach is inefficient, as frequent LRM calls introduce a high computational overhead, and is unreliable, since the LRM as a judge is prone to errors. In this paper, we systematically characterize the capability boundaries of SRMs and identify three common types of reasoning risks: (1) path divergence, where SRMs lack the strategic ability to construct an initial plan, causing reasoning to deviate from the most probable path; (2) cognitive overload, where SRMs fail to solve particularly difficult steps; and (3) recovery inability, where SRMs lack robust self-reflection and error correction mechanisms. To address these challenges, we propose TrigReason, a trigger-based collaborative reasoning framework that replaces continuous polling with selective intervention. TrigReason delegates most reasoning to the SRM and activates LRM intervention only when necessary—during initial strategic planning (strategic priming trigger), upon detecting extraordinary overconfidence (cognitive offload trigger), or when reasoning falls into unproductive loops (intervention request trigger). We show that TrigReason enables more reliable and efficient collaboration between small and large reasoning models, with broad practical application. Under edge–cloud conditions, TrigReason reduces latency by 43.9% and API cost by 73.3% compared to SpecReason.

## 1 Introduction

Large Reasoning Models (LRMs) (OpenAI, 2024; DeepSeek-AI, 2025) have recently emerged as a powerful paradigm for tackling complex problem by leveraging extended chains of thought (CoT) (Wei et al., 2022; Yao et al., 2024a;b) during inference. Unlike standard large language models (LLMs) that directly generate output tokens, LRMs performs an internal reasoning process by generating a sequence of thinking tokens, which break down the input question into intermediate reasoning steps prior to producing the final answer. This structured reasoning behavior enables state-of-the-art performance across diverse domains such as mathematical reasoning (Qwen Team, 2025a), code generation (Ahmad et al., 2025), and agent (Kimi Team, 2025). However, this enhanced reasoning capacity comes at a significant cost: the autoregressive generation of long CoT sequences, often spanning thousands of thinking tokens, leads to prolonged response delays. This limitation has driven recent research into accelerating LRM inference.

Previous approaches to reasoning efficiency have primarily focused on refining the effective density of CoT to mitigate redundant or excessive reasoning. Among these, reinforcement learning with a length penalty is widely adopted to encourage concise and effective reasoning trajectories (Luo et al., 2025; Yang et al., 2025). Alternative methods explore supervised fine-tuning using variable-length CoT data to promote efficient inference (Xia et al., 2025; Kang et al., 2024; Ma et al., 2025). Moreover, prompt engineering also have been proposed to guide models toward more streamlined reasoning through carefully designed input prompts (Wu et al., 2025; Xu et al., 2025). Although these approaches enhance inference efficiency, they typically impose a reduced token budget for reasoning, which may lead to skipping critical logical steps or preventing necessary self-correction in

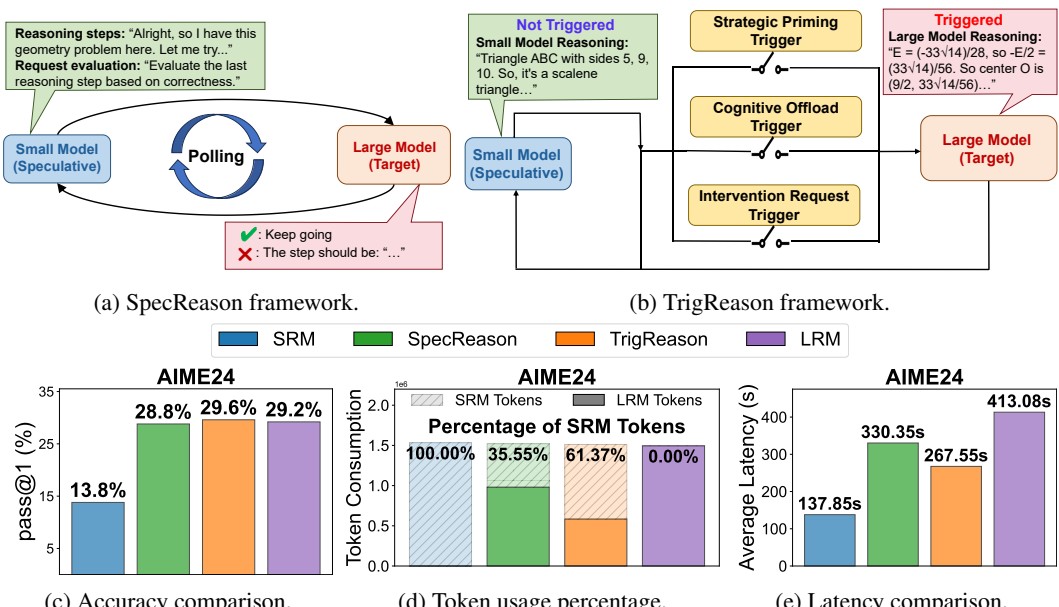

Figure 1: Overview of the reasoning frameworks and performance evaluation between SpecReason and TrigReason. The evaluation is on AIME24 benchmark using DeepSeek-R1-1.5B as SRM and QwQ-32B as LRM

the reasoning process. A separate strand of work aims to develop small language models with strong reasoning capabilities (Dang & Ngo, 2025). However, these methods may also suffer performance degradation due to the limited capabilities of small models.

Recently, SpecReason (Pan et al., 2025) observes that many LRM reasoning steps can be semantically covered by small reasoning model (SRM), and proposes a speculative paradigm that verifies SRM-generated steps via an LRM-as-a-Judge mechanism. While SpecReason can effectively reduce latency, it faces two key limitations. First, the LRM's judgment is often unreliable(§2.1). Model judgment is inherently subjective due to behavior biases ingrained during training. Besides, evaluating individual reasoning steps is challenging, as the full chain of thought remains incomplete. Second, as illustrated in Figure 1a, its polling-based design, where the LRM is invoked at every reasoning step to validate the SRM's output regardless of the complexity, resulting in significant overhead, especially in edge-cloud collaboration (§2.2). These limitations lead to inefficient speculative inference, as excessive LRM intervention results in the final output being dominated by LRM-generated corrections rather than SRM reasoning.

These inefficiencies originate from an incomplete understanding of when and why SRM fails. Existing methods resort to frequent and blind verification, sacrificing efficiency for effectiveness. In this paper, we first characterize the capability boundaries of the SRM to identify the most common reasoning errors: path divergence risk, cognitive overload risk, and recovery inability risk (§3.1). Based on this analysis, we propose **TrigReason**, a event-triggered collaborative reasoning framework that shifts LRM correction from polling to selective intervention. As shown in Figure 1b, instead of continuous verification, TrigReason allows the SRM to reason autonomously until one of three purpose-designed triggers fires: (1) a strategic priming step from the LRM at the start, (2) a cognitive offload trigger when confidence becomes extraordinary, or (3) an intervention request when the SRM detects stagnant reasoning loops. As shown in Figure 1c, 1d and 1e, this shift enables TrigReason to significantly increase the proportion of tokens generated by SRM (from 35.55% to 61.37% of total token consumption) while maintaining accuracy and substantially reducing end-to-end latency.

We evaluate TrigReason extensively on three challenging reasoning benchmarks, AIME24 (AIME, 2024), AIME25 (AIME, 2025), and GPQA Diamond (Rein et al., 2024), across diverse SRM-LRM combinations. Results show that TrigReason maintains accuracy compared with both the full LRM and SpecReason, while utilizing $1.70\times$ to $4.79\times$ more SRM-generated tokens than SpecReason, indicating significantly higher reasoning steps offloading efficiency. Under edge-cloud collaboration

scenarios, TrigReason achieves reduction of 43.9% in latency and 73.3% in API cost compared to SpecReason. These results demonstrate that TrigReason establishes a more effective paradigm for collaborative reasoning between small and large models, achieving significant improvements in inference efficiency without compromising accuracy.

## 2 MOTIVATION

Recent advances in speculative reasoning have shown that collaboration between SRM and LRM can accelerate inference without sacrificing solution quality. SpecReason (Pan et al., 2025) exemplifies this progress by adopting an LRM-as-a-Judge framework, where the LRM is prompted to score each reasoning step generated by the SRM, determining whether to accept or reject it. In this section, we explore two key limitations hindering its practical effectiveness:

**Unreliable LRM judgment**: (1) the inherent biases of each model, which make the judge prone to subjectivity rather than serving as an objective detector of errors; and (2) the difficulty of assessing intermediate reasoning steps when chains of thought are not yet fully formed, thus often unclear.

**Inefficiency of polling-based execution**: the step-level granularity of LRM invocation leads to frequent communication and computation overhead, especially in edge-cloud collaboration.

These limitations undermine intended acceleration benefits of speculative reasoning, as the majority of the final output is derived from the corrections of LRM.

### 2.1 UNRELIABILITY OF LRM JUDGMENT

While SpecReason advances speculative inference efficiency in reasoning acceleration, its LRM-as-a-Judge mechanism suffers from a critical flaw, as LRM fails to reliably validate the correctness of reasoning steps from SRM.

Due to inherent biases in model training, LRM judgments are often preference-driven, leading to subjective judgments for the same content across different models. We evaluate identical reasoning trajectories using four different LRMs. As shown in Figure 2a, the LRMs assign widely divergent scores to the same trajectory (from 1.87 to 8.93). This polarization indicates that LRM judgments are heavily influenced by model-specific priors, rather than objective reasoning quality.

Furthermore, to assess the difficulty of verifying intermediate reasoning steps in incomplete chains of thought, we sample reasoning trajectories from the AIME24 dataset. We categorize these trajectories into three types (defined below), and evaluate SpecReason to quantify the unreliability verification, using QwQ-32B as the LRM and DeepSeek-R1-1.5B as the SRM:

- **SpecReason**: trajectories of SpecReason with mixed steps from SRM and LRM;

- **SRM-Correct**: trajectories from the SRM that yield correct final answers;

- **LRM-Own**: trajectories generated independently by the LRM itself.

We follow SpecReason's experimental setup: the LRM assigns scores in the range [0, 9], with a threshold of 7 for acceptance, and each question is evaluated over 16 runs to compute the average rejection rate across reasoning trajectories. Figure 2b presents results on three questions where the SRM can correctly solve, as the remaining results are shown in Appendix A. The results reveal that the LRM rejects **50.1% to 80.9%** of correct and valid reasoning steps generated by the SRM. More surprisingly, the LRM even rejects up to **63.7%** of its own generation.

These results indicate that the LRM-as-a-Judge paradigm is unreliable in verifying reasoning steps. Due to discrepancies of model inherent biases and the difficulty of assessing intermediate reasoning steps, the LRM is prone to regenerate the correct draft of SRM that differ only in phrasing or reasoning path. This unreliable judgment forces excessive LRM intervention to ensure solution quality, significantly undermining the efficiency of speculative reasoning.

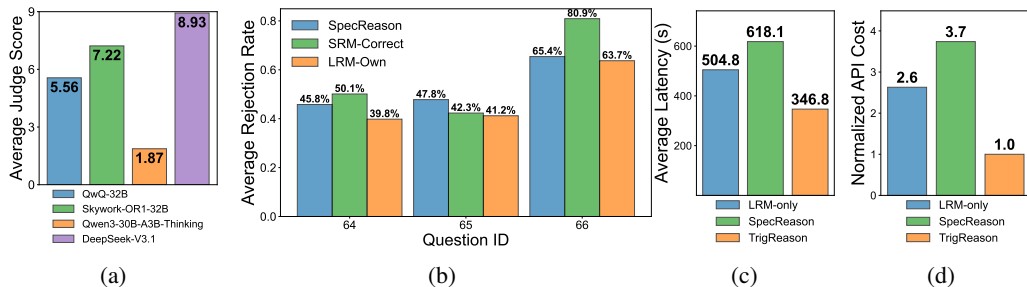

(a)  (b)  (c)  (d)

Figure 2: (a) Average judge scores of same trajectories from four different LRM, showing extreme inter-judge inconsistency. (b) Average rejection rate on three reasoning trajectories. Even for correct steps and LRM-own generation, LRM judgment still shows high-level rejection rate. (c) and (d) are comparison of average latency and API cost in edge-cloud collaborative reasoning, respectively.

## 2.2 INEFFICIENCY OF POLLING-BASED EXECUTION

SpecReason adopts polling-based execution that requires the LRM to intervene at every reasoning step, ignoring both step complexity and the SRM's internal confidence. Frequent LRM calls incur substantial overhead, and also diminish the expected efficiency gains of speculative reasoning.

In edge-cloud setups, a representative deployment for speculative reasoning, the SRM executes on resource-constrained edge devices while the LRM operates in the cloud. Frequent polling under this architecture induces significant **network round-trips and API costs**, exacerbating system-level inefficiencies. We implement a edge-cloud deployment to quantify the effect, as the SRM (DeepSeek-R1-1.5B) runs locally and the LRM is accessed via DeepSeek API. As shown in Figure 2c and 2d, even compared to LRM-only execution, SpecReason exhibits lower efficiency, with a 22.44% increase in latency and a 42.31% higher API cost.

## 3 METHOD

To address the shortcomings of polling-based LRM verification, we propose TrigReason, guided by two key ideas: (1) studying how SRM fails in order to identify the most common reasoning risks, thereby enabling more objective targeting and reducing blind reliance on LRM judges; and (2) triggering LRM for speculative reasoning correction based on event signals rather than at every step (polling), which reduces the number of LRM calls and significantly lowers latency. Allowing SRM to generate the reasoning chain to some extent before intervention also makes the reasoning path more explicit, enabling LRM to deliver more effective corrections.

### 3.1 CHARACTERIZATION OF SRM REASONING RISKS

The limitations of current speculative reasoning originate from an insufficient understanding of **when and why** SRM fails, resulting in an inability to distinguish between harmless reasoning variations and high-risk steps. By characterizing the capability boundaries of the SRM, LRM intervention can be reserved only for critical steps, avoiding excessive verification and missed interventions. To address this issue, we conduct a systematic analysis of reasoning trajectories generated by SRM and identify three core failure modes that cause distinct capability gap between SRM and LRM: path divergence risk, cognitive overload risk, and recovery inability risk.

To identify the critical steps and risk patterns, we compared correct and incorrect reasoning trajectories through different scaled reasoning models on the AIME24 datasets. Figure 3 shows three typical risk patterns, which characterize the capability gap between different model scales. Examples of the three risks are shown in Appendix B.

**(1) Path Divergence Risk:** occurs at the beginning of reasoning process, representing a fault-oriented solution branch. Unlike factual hallucinations or arithmetic mistakes, it arises from the SRM's failure to decompose the problem or anticipate the implications of alternative approaches.

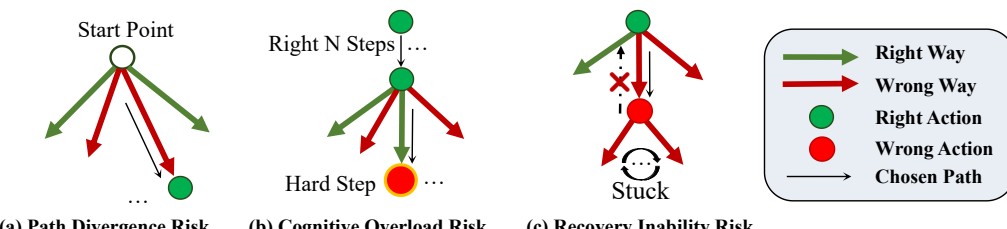

Figure 3: Illustration of three typical risk patterns reflecting the SRM-LRM capability gap.

As the study case shown in Appendix B, SRMs often jump to computation or apply familiar but unsuitable methods, whereas larger models first analyze problem structure and plan strategically. The observation highlights the lack of strategic foresight in smaller models during initial planning.

**Insight:** let the LRM generate initial reasoning steps for strategic planning or problem decomposition, enabling the SRM to efficiently execute downstream steps under a validated reasoning path.

**(2) Cognitive Overload Risk:** occurs in a subset of specific steps that demand high cognitive load, such as complex arithmetic computations requiring the retention of numerous intermediate states (e.g., multi-step fraction simplification, symbolic manipulation, or multi-hop logical inference).

As the study case shown in Appendix B, although relatively infrequent, these errors are highly consequential, leading cascading failures in subsequent reasoning steps. Yet for routine reasoning (e.g., interpretation, sequencing, and simple calculation), smaller models are reliable. Therefore, the key lies in identifying a insightful signal to detect cognitive overload in the SRM.

We analyze 160 reasoning trajectories from SRMs on 10 problems with significant SRM–LRM performance gaps. Among the 93 trajectories containing clear SRM incorrect reasoning steps, 94.6% steps exhibit overconfident steps (i.e., over 85% of tokens have perplexity $< 1.05$), compared to only 38.1% overconfidence steps across all steps (see Appendix C for details). This pattern shows SRM failure is often preceded by abnormally low token-level perplexity, indicating overconfident and deterministic generation. The overconfidence is not a sign of capability, but rather a symptom of mechanical pattern completion under cognitive overload.

**Insight:** SRMs are limited not by reasoning ability throughout the process, but by cognitive capacity. This pattern motivates a light-touch intervention strategy: leveraging overconfidence as a signal of cognitive overload to trigger LRM assistance, keeping the SRM on a correct reasoning trajectory.

**(3) Recovery Inability Risk:** occurs when minor errors or ambiguous interpretations lead the SRM to deviate from the correct path, resulting in increasingly incoherent reasoning. In contrast, LRM can implicitly reflect, detect anomalies, and backtrack to correct its approach.

As the study case shown in Appendix B, SRMs lack the ability to detect deviations or initiate self-correction, causing them to persist on erroneous paths and eventually stagnate.

**Insight:** let LRM to reflect and re-guide the path-enabling corrective when signs of stagnation or contradiction are detected in the SRM's reasoning trajectory.

### 3.2 EVENT-TRIGGERED LRM INTERVENTION

Based on the observations in §3.1, we propose TrigReason, a trigger-based framework for collaborative SRM-LRM execution. TrigReason introduces three targeted triggers to address critical failure risks in SRM reasoning: **strategic priming trigger**, **cognitive offload trigger**, and **intervention request trigger**. Owing to sparse yet crucial LRM intervention, TrigReason ensures high answer quality while enabling efficient and low-cost collaborative reasoning, as shown in Figure 2c and 2d.

#### 3.2.1 STRATEGIC PRIMING TRIGGER

The Strategic Priming Trigger is designed to address the Path Divergence Risk. By decoupling strategic planning from step-by-step execution, TrigReason uses the LRM to perform initial reasoning, ensuring the SRM begins on a valid and coherent trajectory.

Specifically, given an input question $x$, we first sample the first $n$ reasoning steps from the LRM $L$:

$$y_{1:n} \sim p_L(y_{1:n} \mid x), \tag{1}$$

where $p_L$ denotes the conditional distribution of the LRM, and $n$ is a pre-defined priming steps. After this priming phase, control is transferred to the SRM $S$, which continues the reasoning chain:

$$y_t \sim p_S(y_t \mid y_{<t}, x), \quad \text{for } t > n. \tag{2}$$

### 3.2.2 COGNITIVE OFFLOAD TRIGGER

The Cognitive Offload Trigger aims to address the Cognitive Overload Risk. TrigReason leverages the extraordinary overconfidence of SRM as an early warning signal to trigger LRM intervention at critical junctures.

To quantify this behavior, we define the token-level perplexity at position $t$ as:

$$PPL(t) = \exp\left(-\log p_S(y_t \mid y_{<t}, x)\right), \tag{3}$$

where $p_S(y_t \mid y_{<t}, x)$ is the probability assigned by the SRM to token $y_t$, given the prefix $y_{<t}$ and input $x$. For a given reasoning step $s$, let $T_s$ denote the set of token positions in $s$. We compute the low-perplexity ratio $r_s$ as the fraction of tokens in $s$ with perplexity below a threshold $\tau$:

$$r_s = \frac{1}{|T_s|} \sum_{t \in T_s} \mathbf{1}\left[PPL(t) < \tau\right], \tag{4}$$

where $\tau$ is a sensitivity threshold and $\mathbf{1}[\cdot]$ is the indicator function, equal to 1 if the condition is true, and 0 otherwise. The Cognitive Offload Trigger fires when $r_s > \rho$, where $\rho$ is a coverage threshold:

$$\text{Trig}_{\text{cognitive}}(s) = \mathbf{1}\left[r_s > \rho\right]. \tag{5}$$

Upon activation, the current step $s$ is regenerated by the LRM:

$$y_s \sim p_L(\cdot \mid y_{<s}, x). \tag{6}$$

### 3.2.3 INTERVENTION REQUEST TRIGGER

The Intervention Request Trigger aims to mitigate the Recovery Inability Risk. TrigReason monitors for linguistic markers of reasoning stagnation, and invokes the LRM to realign the reasoning path upon detection. Obesevation indicates that, SRM often generates distinctive hesitation patterns (e.g., "wait", "hmm", "alternatively"), which reflects an implicit recognition of difficult reasoning steps.

We define a finite set $\mathcal{H}$ of hesitation words for detection (Appendix D includes the complete list):

$$\mathcal{H} = \{\text{wait}, \text{hmm}, \text{alternatively}, \ldots\}. \tag{7}$$

At each reasoning step $s$, we determine whether the generation contains at least one token from $\mathcal{H}$:

$$h_s = \mathbf{1}\left[\exists\, y_t \in y_s \text{ such that } y_t \in \mathcal{H}\right]. \tag{8}$$

The Intervention Request Trigger fires when hesitation is observed in $k$ consecutive steps:

$$\text{Trig}_{\text{intervention}} = \mathbf{1}\left[\sum_{i=0}^{k-1} h_{s-i} = k\right], \tag{9}$$

Upon activation, the system transfers control to the LRM for the next $m$ steps:

$$y_{s+1:s+m} \sim p_L(\cdot \mid y_{\leq s}, x), \tag{10}$$

then LRM is able to assess the current state, identify inconsistencies, and recorrect reasoning path. After $m$ steps, control returns to the SRM. The main algorithm of TrigReason is shown in Appendix E Algorithm 1 and the theoretical Characterization of reliability is discussed in Appendix G.

## 4 EXPERIMENTS

### 4.1 SETUP

**Models.** **LRM**: QwQ-32B (Qwen Team, 2025a) (32B dense model) and Qwen3-30B-A3B-Thinking-2507 (Qwen Team, 2025b) (30B MoE model, 3B active). **SRM**: DeepSeek-R1-1.5B (DeepSeek-AI, 2025) and Qwen3-0.6B (Qwen Team, 2025b). Both models are equipped with CoT reasoning capabilities. We conduct experiments across four SRM-LRM pairings to evaluate the generalization of TrigReason under diverse model architectures and scales. In the edge-cloud deployment setting, the LRM (DeepSeek-V3.1, 671B MoE) is accessed via DeepSeek API.

**Datasets. AIME24** (AIME, 2024) and **AIME25** (AIME, 2025) are high-school math competition problems requiring multi-step algebraic and combinatorial reasoning. **GPQA Diamond** (Rein et al., 2024) is a graduate-level multiple-choice question set that covers advanced topics in physics, chemistry and biology, known for its high factual and logical complexity.

**Evaluation Metrics.** (1) **Accuracy**: following prior work (Pan et al., 2025), we use **pass@1** with k = 16. Specifically, 16 responses are sampled for each question at temperature = 0.6, and the final accuracy is calculated as the average accuracy for every response. (2) **Efficiency**: as the total token consumption across methods is similar, we utilize the ratio of tokens generated by the SRM to the total reasoning tokens as a robust efficiency metric, donated as **SMT percentage**. We exclude latency from evaluation, as it is highly sensitive to hardware, system load, and scheduling variability, which could confound cross-method comparisons.

The method performance is visualized through the **Accuracy-Efficiency** plane, where the *x* and *y* axis represent *SMT percentage* and *pass@1*, respectively. Closer to the top-right performs better.

**Baselines.** (1) **SpecReason** (Pan et al., 2025), a polling-based collaborative method. (2) standalone reasoning framework using **only the SRM** or **only the LRM**.

**Implementation Details.** All experiments are conducted on 8 NVIDIA RTX 4090 GPUs using SGLang v0.4.9 (Zheng et al., 2023) as the inference engine, with prefix caching and tensor parallelism (degree 4) enabled. Unless otherwise stated, generation uses temperature = 0.6 and top_p = 0.95. The default token budget is 8192 tokens; for the impact of thinking budget analysis (Appendix F), we evaluate budgets ranging from 2K to 32K. For TrigReason parameters, we set the priming step count $n = 20$ and rectification steps $m = 1$. The cognitive overload threshold $\rho$ is set to 0.85 for DeepSeek-R1-1.5B and 0.75 for Qwen3-0.6B with $\tau = 0.85$. The rationale for these hyperparameter choices is justified through ablation studies in (§4.4).

### 4.2 MAIN RESULTS

We evaluate TrigReason on AIME24, AIME25, and GPQA Diamond across four SRM-LRM combinations, comparing against vanilla LRM/SRM baselines and SpecReason. The results are shown in Figure 4.

**Stable Accuracy.** Despite offloading substantial reasoning to the SRM, TrigReason consistently matches or even exceeds LRM performance. On average, it achieves 105.8% (AIME24), 104.7% (AIME25), and 99.6% (GPQA Diamond) of the LRM's accuracy across model pairs, with individual configurations (Qwen3-0.6B + Qwen3-30B-A3B-Thinking- 2507 on AIME24) reaching up to 119.3%. In several cases, TrigReason surpasses the LRM baseline, suggesting that trigger-based intervention can yield robust and effective reasoning trajectories.

**Higher Efficiency.** While matching SpecReason in accuracy, TrigReason achieves significantly greater efficiency. On average, TrigReason utilizes $1.70 \times -4.79 \times$ more SRM tokens than SpecReason across benchmarks. Specifically, the SMT Percentage increases by $1.70\times$, $4.79\times$, $1.88\times$, and $3.94\times$ across the four combinations. This substantial gain indicates that TrigReason's mechanism more effectively identifies and accepts valid reasoning steps.

In general, the results show that TrigReason achieves accuracy on par with full LRM and SpecReason, while significantly improving efficiency through increased SRM utilization and less LRM calls. The evaluation results indicate that TrigReason achieves a superior efficiency-accuracy trade-off, representing a clear advancement in step-level speculative reasoning for collaborative inference.

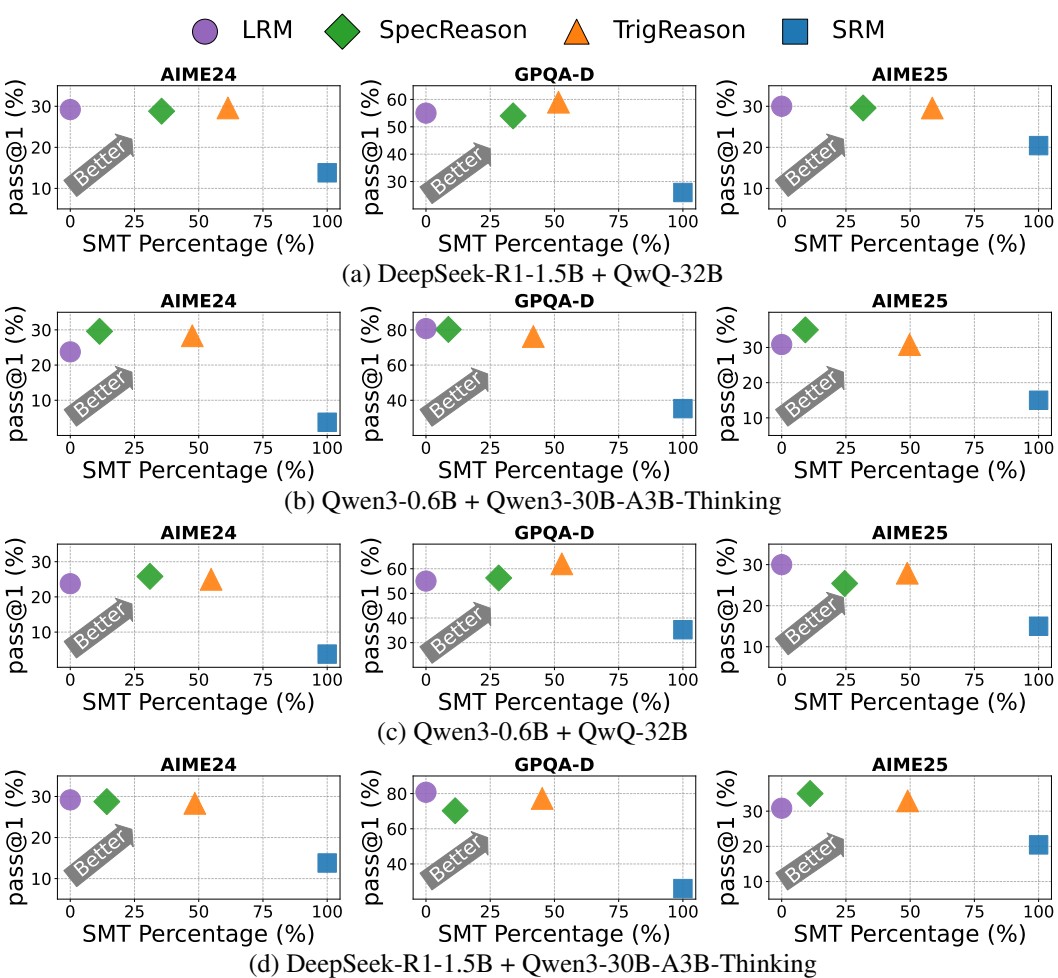

Figure 4: Performance comparison across benchmarks and model combinations. The vertical axis shows accuracy (higher is better), and the horizontal axis shows the percentage of tokens generated by the SRM (SMT Percentage), reflecting reasoning efficiency (higher is more efficient). Methods closer to the top-right corner achieve better accuracy with greater computational offloading to the SRM, indicating a favorable trade-off between performance and efficiency.

### 4.3 EVALUATION IN EDGE-CLOUD COLLABORATION

To assess TrigReason in realistic deployment scenarios, we simulate an edge-cloud setup: the SRM (DeepSeek-R1-1.5B) runs locally, while the LRM is accessed remotely via the DeepSeek API (DeepSeek API, 2025), which internally uses DeepSeek-V3.1. Latency and API cost are reported in Figure 2c and Figure 2d; here, we present accuracy results on AIME24 in Figure 5.

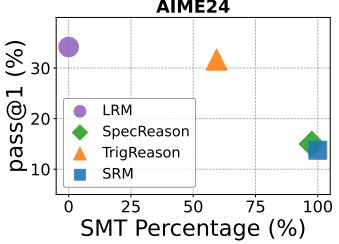

Figure 5: Accuracy on AIME24 in an edge-cloud setup.

TrigReason successfully offloads up to 59.4% of reasoning tokens to the SRM—requiring the LRM to generate only 40.6%, with just a 2.49% absolute accuracy drop compared to full LRM execution. In contrast, SpecReason suffers from degraded accuracy despite high SRM token usage, due to unreliable verification by the LRM, as analyzed in Section 2.1. We observe that when acting as verifier, DeepSeek-V3.1 often assigns high scores to semantically weak or incomplete SRM-generated steps, likely influenced by its own inductive biases in reasoning style. This leads to acceptance of invalid speculative steps, enabling error propagation and ultimately compromising solution correctness.

## 4.4 ABLATION STUDY

To evaluate the contribution of each component in TrigReason, we conduct ablation studies on the three proposed triggers and their associated hyperparameters: Strategic Priming, Cognitive Offload, and Intervention Request. All experiments are conducted on AIME24.

**Cognitive Overload Threshold** $\rho$. We analyze $\rho$, which governs activation of the *Cognitive Offload Trigger*. Lower $\rho$ prompts earlier LRM intervention; $\rho = 1$ disables the trigger entirely. We fix $n = 20$, $m = 1$, and use two representative model pairs: DeepSeek-R1-1.5B + QwQ-32B and Qwen3-0.6B + Qwen3-30B-A3B-Thinking.

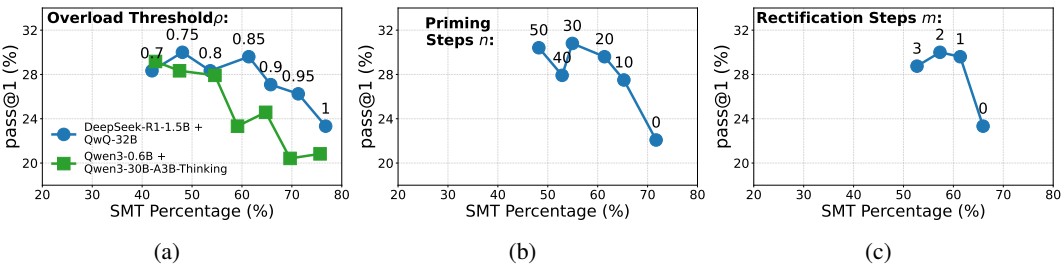

(a)                                    (b)                                    (c)

Figure 6: Ablation studies on TrigReason's three triggers and their hyperparameters. (a) Impact of the cognitive overload threshold $\rho$ on accuracy and SRM token percentage across two model pairs. (b) Effect of varying the number of priming steps $n$ on accuracy and SRM token usage. (c) Performance and efficiency under different rectification step counts $m$.

As shown in Figure 6a, disabling the Cognitive Offload Trigger ($\rho = 1$) causes a significant accuracy drop, confirming its critical role in preventing error accumulation when the SRM exceeds its capacity. Crucially, the optimal $\rho$ is model-dependent: DeepSeek-R1-1.5B + QwQ-32B pair achieves peak performance at $\rho = 0.85$, while Qwen3-0.6B + Qwen3-30B-A3B-Thinking pair performs more stably at $\rho = 0.75$. This reflects intrinsic SRM differences in average perplexity and reasoning reliability. While higher $\rho$ improves accuracy, it reduces SMT percentage, trading off efficiency. Thus, $\rho$ acts as a tunable knob balancing accuracy and efficiency based on the specific SRM-LRM pair.

**Priming Steps** $n$. We vary $n$ from 0 to 50 (with $\rho = 0.85$ and $m = 1$) to assess the *Strategic Priming Trigger*, which enables LRM-provided planning before SRM execution.

Figure 6b shows that reducing $n$ from 20 to 0 incurs a 25.4% absolute accuracy drop, underscoring the importance of strategic guidance in enabling autonomous SRM reasoning. However, increasing $n$ beyond 30 yields diminishing returns and degrades efficiency. This indicates that early strategic guidance is critical, while excessive priming wastes LRM capacity on execution.

**Rectification Steps** $m$. We evaluate the *Intervention Request Trigger* by varying $m$, the number of LRM-generated steps after detecting stagnant reasoning loops. We fix $\rho = 0.85$, $n = 20$.

Figure 6c shows that $m = 1$ already recovers most of the performance gap, with marginal gains from $m = 2$ or $m = 3$. This suggests that the LRM's corrective capability is highly concentrated: a single high-quality step often suffices to realign the reasoning path. Larger $m$ unnecessarily increases LRM usage and reduces efficiency, making $m = 1$ the optimal trade-off in practice.

## 5 CONCLUSION

We introduced TrigReason, a trigger-based framework for efficient large-small reasoning model collaboration. By replacing polling with risk-aware, selective intervention, TrigReason enables autonomous SRM reasoning while invoking LRMs only when neccessary. Across mathematical and knowledge-intensive benchmarks, TrigReason sustains LRM-level accuracy while offloading up to 59.4% of tokens to SRMs, reducing latency by 43.9% and API cost by 73.3% in edge-cloud setups. Our work shows that intelligent triggering, informed by failure analysis, enables an efficient and reliable path to scalable reasoning systems.

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

## A    DETAILED ANALYSIS OF LRM JUDGMENT

To further investigate the unreliability of the LRM-as-a-Judge mechanism in SpecReason, we conduct a fine-grained analysis of rejection behavior on the AIME24 benchmark. Specifically, we compare the step-level rejection rates of two conditions across all 30 questions: (1) the original SpecReason setup, where the LRM judges SRM-generated reasoning steps, and (2) an LRM-own control, where the LRM judges its own reasoning trajectory generated during full LRM execution.

Figure 7 presents the per-question rejection rates for both settings. Despite the semantic correctness of its own reasoning path, the LRM rejects its own steps at a rate comparable to that of SRM-generated steps—indicating inconsistent and preference-driven judgment. On average, the LRM rejects 53.4% of its own reasoning steps, only slightly lower than the 56.8% rejection rate for SRM steps. This high self-rejection rate suggests that the LRM's scoring mechanism is not grounded in logical validity, but rather in stylistic or strategic preferences.

This finding strongly supports our claim in Section 2.1: using the LRM as a judge introduces inherent unreliability, as it cannot reliably distinguish between valid reasoning variations and genuinely erroneous steps. Consequently, SpecReason may reject correct SRM reasoning or accept flawed ones based on superficial alignment, undermining both efficiency and correctness.

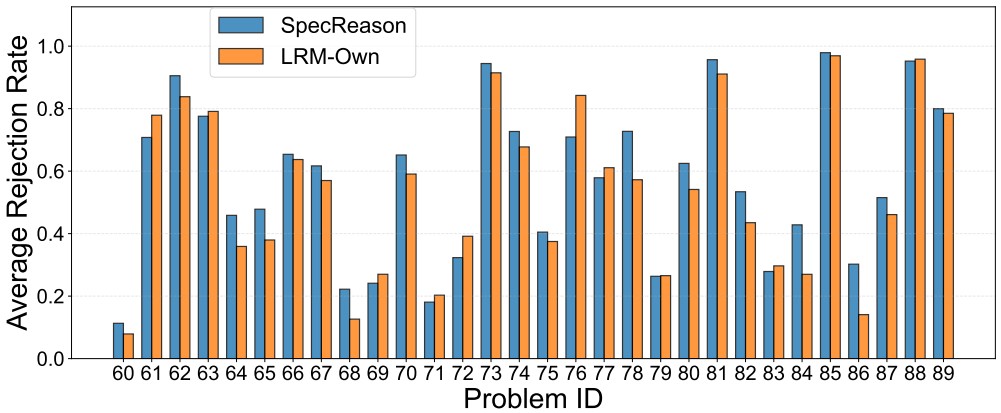

Figure 7: Per-question rejection rates of SRM-generated steps (SpecReason) vs. LRM-generated steps (LRM-own) on AIME24. The LRM frequently rejects its own valid reasoning, revealing the inconsistency of its judgment.

## B    CASE STUDIES OF THREE RISK PATTERNS

To provide intuitive illustrations of the three key risk patterns identified in Section 3.1, we present visual case studies on representative problems from the AIME24 benchmark.

- Figure 8 demonstrates the Path Divergence Risk, where the SRM makes suboptimal procedural choices that lead to intractable computation, while the LRM adopts a more strategic formulation.
- Figure 9 illustrates Cognitive Overload Risk: the SRM performs correctly for hundreds of steps but fails at a critical late-stage computation due to arithmetic or attentional lapse.
- Figure 10 showcases Recovery Inability Risk, where the SRM enters a loop of indecisive reasoning after hitting a bottleneck, failing to self-correct or switch strategies.

## Path Divergence Risk

**Question (AIME24-61):**
Let $ABC$ be a triangle inscribed in circle $\omega$. Let the tangents to $\omega$ at $B$ and $C$ intersect at point $D$, and let $\overline{AD}$ intersect $\omega$ at $P$. If $AB=5$, $BC=9$, and $AC=10$, $AP$ can be written as the form $\frac{m}{n}$, where $m$ and $n$ are relatively prime integers. Find $m + n$.

**SRM Answers:**
Alright, so I have this geometry problem here. Let me try to visualize it first. There's a triangle ABC inscribed in a circle $\omega$. The tangents at points B and C intersect at point D. Then, the line AD intersects the circle $\omega$ again at point P. I know the lengths of the sides: AB is 5, BC is 9, and AC is 10. I need to find AP expressed as a fraction m/n where m and n are coprime, and then compute m + n.

Hmm, okay. Let me recall some geometry concepts that might help here. Since AD intersects the circle again at P, maybe I can use power of a point or some properties of cyclic quadrilaterals. Also, since D is the intersection of the tangents at B and C, D must be the ex-pole or something related to the circumcircle.

Wait, tangents from a common external point are equal in length. So, DB equals DC. That might be useful. Also, since ABC is inscribed in $\omega$, maybe I can use properties of cyclic quadrilaterals or intersecting chords.

I remember that the power of point D with respect to $\omega$ is equal to DB², since DB is tangent to $\omega$ at B. Similarly, it's also equal to DC² because DC is tangent at C. So, DB = DC.

Now, since AD intersects $\omega$ again at P, perhaps I can apply the power of point A with respect to $\omega$. The power of A would be equal to AP * AD. Also, the power of A can be expressed as AB * AC because A is outside the circle and AB and AC are the secant segments from A to the circle.

Wait, is that right? Let me think. The power of a point outside a circle is equal to the product of the lengths of the segments from the point to the points of intersection with the circle. So, for point A, the power with respect to $\omega$ is AB * AC because AB and AC are the two secant segments from A to points B and C on the circle. But actually, in this case, A is connected to P, which is another point on the circle. So, AP * AD should equal AB * AC.
……
Wait, maybe using Ceva's theorem or Menelaus' theorem. But without knowing certain angles or lengths, it's difficult.

Wait, maybe I can compute the coordinates. Let me set up coordinate system.

**Let me place triangle ABC in coordinate plane. Let me assign coordinates to points A, B, C. Let me let point A be at (0, 0), point B at (5, 0), and point C somewhere in the plane. Wait, but I don't know the coordinates yet.**

**Comment:**
While placing $A$ at $(0,0)$ and $B$ at $(5,0)$ is mathematically valid, this coordinate system leads to significantly more complex algebraic expressions in subsequent steps—such as solving for the circumcircle, computing tangent lines, and finding intersection points—making it extremely difficult to reach the correct solution within the token budget. This illustrates a key aspect of path divergence risk: even when an SRM follows a logically sound high-level strategy, suboptimal procedural choices (e.g., coordinate placement) can lead to intractable computational paths. In contrast, we observe that LRMs frequently adopt a more strategic setup—placing $B$ at $(0,0)$ and $C$ at $(9,0)$—which aligns the base $BC$ with the x-axis and leverages the known side length $BC = 9$ as a natural coordinate anchor. This choice dramatically simplifies symmetry and distance calculations, reducing reasoning complexity and increasing the likelihood of success. The SRM's failure to anticipate such downstream computational consequences highlights its limited global planning capability, motivating the need for strategic priming (e.g., via the Strategic Priming Trigger in TrigReason) to guide early critical decisions.

Figure 8: Case study of Path Divergence Risk.

702
703
704
705
706
707
708
709
710
711
712
713
714
715
716
717
718
719
720
721
722
723
724
725
726
727
728
729
730
731
732
733
734
735
736
737
738
739
740
741
742
743
744
745
746
747
748
749
750
751
752
753
754
755

## *Cognitive Overload Risk*

**Question (AIME24-61):**
Let $ABC$ be a triangle inscribed in circle $\omega$. Let the tangents to $\omega$ at $B$ and $C$ intersect at point $D$, and let $\overline{AD}$ intersect $\omega$ at $P$. If $AB=5$, $BC=9$, and $AC=10$, $AP$ can be written as the form $\frac{m}{n}$, where $m$ and $n$ are relatively prime integers. Find $m + n$.

**SRM Answers1:**

......
Let me place triangle ABC in the coordinate plane. Let me assign coordinates to the points A, B, and C. Let me assume point B is at (0,0), point C is at (9,0), since BC is 9 units. Then, point A is somewhere in the plane such that AB = 5 and AC = 10.
Let's set B at (0,0). Let's set C at (9,0). Let's let A be at (x,y). Then, the distance from A to B is 5, so sqrt((x)^2 + (y)^2) = 5, so x^2 + y^2 = 25.
......
(1/3)^2 + y^2 = 25 => 1/9 + y^2 = 25 => y^2 = 25 - 1/9 = 225/9 - 1/9 = 224/9 => y = sqrt(224/9) = (sqrt(224))/3 = (2*sqrt(56))/3 = (2*2*sqrt(14))/3 = 4*sqrt(14)/3.
So, coordinates of A are (1/3, 4√14/3).
......
Thus, the equation of circle ω is:
x^2 + y^2 - 9x - (33√14/28)y = 0.
......
So, the tangent at C is 84x - 11√14 y - 756 = 0.
Now, we have the equations of the two tangents:
Tangent at B: 84x + 11√14 y = 0.
Tangent at C: 84x - 11√14 y - 756 = 0.
We can solve these two equations to find the coordinates of point D.
......
Therefore, power of D: DB^2 = DA * DP => (225/22)^2 = (325/22) * DP.
So, solve for DP:
50625 /22 /325 = (50625 /22) * (1/325) = (50625 / (22*325)).
**Simplify numerator and denominator:**
**50625 ÷25=2025, 22*325=7325.**
**So, 2025 /7325.**
......
================================================================================
**SRM Answers2:**

......
**AP = sqrt[(1100² + (1000√14)²)/507²] = sqrt[(1,210,000 + 1,400,000)/507²] = sqrt[(2,610,000)/507²] = sqrt[2,610,000]/507.**
......

**Comment:**
We observe a hallmark of cognitive overload risk: the SRM performs correctly over hundreds of reasoning steps but fails at a single, critical computation late in the process—often due to arithmetic fatigue or attention drift under long context. The two examples above illustrate this phenomenon starkly. In the first case, after correctly deriving $DP = 50625 / (22 \times 325)$, the SRM attempts to simplify by dividing both numerator and denominator by 25, but erroneously applies the division only to the numerator ($50625 \div 25 = 2025$), leaving the denominator unmodified. This inconsistent operation breaks the equality and invalidates the result. In the second case, the model correctly sets up the expression for $AP$ and is very close to the final correct result, but during the final computation of $(1000\sqrt{14})^2$, it computes $1,400,000$ instead of the correct $14,000,000$—a single-digit error (missing a zero) that nullifies hundreds of prior correct steps. Both failures occur not from flawed logic, but from transient computational lapses under cognitive strain. This supports our design of the Cognitive Offload Trigger, which monitors confidence and irregularity in intermediate computations to proactively offload high-risk steps before such errors occur.

Figure 9: Case study of Cognitive Overload Risk.

### Recovery Inability Risk

**Question (AIME24-61):**
Let $ABC$ be a triangle inscribed in circle $\omega$. Let the tangents to $\omega$ at $B$ and $C$ intersect at point $D$, and let $\overline{AD}$ intersect $\omega$ at $P$. If $AB=5$, $BC=9$, and $AC=10$, $AP$ can be written as the form $\frac{m}{n}$, where $m$ and $n$ are relatively prime integers. Find $m + n$.

**SRM Answers:**
......
Thus, AP= (1,292,735 /1,084,519)*(sqrt(68,324,697)/1188).
Wait, but that seems messy.
Wait, perhaps another approach. Since P lies on AD, and we have the coordinates of D, which is (7441/1188, 481 sqrt(14)/594), and P is beyond D, so P is a scalar multiple of AD beyond D.
Thus, vector AD is D - A= (7441/1188, 481 sqrt(14)/594).
Thus, P= D + k*(D - A), where k>0.
But perhaps this is not helpful.
Alternatively, since we have the coordinates of P as:
x= (1,292,735 /1,084,519)*(7441/1188)= (1,292,735 *7441)/(1,084,519 *1188).
Similarly, y= (1,292,735 /1,084,519)*(481 sqrt(14)/594)= (1,292,735 *481 sqrt(14))/(1,084,519 *594).
But this seems too cumbersome. Maybe we can find the ratio t=1,292,735 /1,084,519.
But 1,292,735 ÷5=258,547.
1,084,519 ÷5=216,903.8, which is not integer. So, maybe t is already in simplest terms.
Alternatively, maybe we can rationalize AP.
But perhaps it's better to recall that AP= t*AD.
......
But we need AP in exact terms.
Alternatively, perhaps we can find AP as the length from A to P, which is beyond D on AD.
But without knowing the exact coordinates, it's difficult.
Alternatively, perhaps we can use power of a point.
Wait, another idea: since A is on the circumcircle, and AD is a secant. So, power of A with respect to ω is zero, as A is on ω.
But perhaps using power of point P with respect to ω.
Power of P: PA * PD = PB * PC.
But since P is on AD, which is a secant, and also, P is on the circumcircle ω.
Wait, no, P is on ω, so PA * PD=0, but that doesn't help.
Alternatively, since P is the intersection of AD with ω, and D is the other intersection point, so PA * PD= power of A with respect to ω, but A is on ω, so power is zero, which again PA * PD=0, which only tells us that either PA=0 or PD=0, which isn't helpful.
Wait, perhaps it's better to consider triangle ABD and power of point.
Wait, point P lies on AD and on ω. So, PA * PD= power of A with respect to ω. But since A is on ω, power is zero, so PA * PD=0. So, either PA=0 or PD=0, which is trivial.
So, that approach doesn't help.
Alternatively, since A is on ω, and AD is a secant through D, then power of D with respect to ω is equal to DA * DP= DB * DC.
Wait, yes, that's the power of a point theorem.
......

**Comment:**
A key strength of advanced reasoning models is their ability to self-reflect, backtrack, and recover from incorrect paths—often through iterative hypothesis testing and strategic redirection. However, we observe that SRMs frequently lack this recovery capability, leading to what we term Recovery Inability Risk. As shown in the example, after hitting a reasoning bottleneck, the SRM begins to generate repetitive, indecisive patterns marked by phrases like"Alternatively,""perhaps," and"Wait", indicating uncertainty and failed hypothesis generation. Rather than backtracking to a valid state or switching to a fundamentally different strategy, the model loops in a state of semantic hesitation, unable to escape the flawed trajectory. In contrast, LRMs typically detect such stagnation and invoke insights to break the deadlock. This fundamental disparity motivates the Intervention Request Trigger in TrigReason, which identifies linguistic and structural markers of stagnation and requests timely LRM intervention to reset and redirect the reasoning process.

Figure 10: Case study of Recovery Inability Risk.

## C  OVERCONFIDENCE AS A SIGN OF COGNITIVE OVERLOAD

We analyze 160 reasoning trajectories from SRMs across 10 problems where SRMs and LRMs exhibit significant performance gaps. In this analysis, we identify 93 trajectories containing clearly incorrect reasoning steps. A striking pattern emerges: these erroneous steps frequently exhibit overconfidence. To quantify this, we compute the proportion of *low-perplexity tokens* (defined as tokens with per-token perplexity $< 1.05$) within each reasoning step. Among the 93 trajectories with identifiable errors, 88 (94.6%) contain steps where over 85% of tokens are low-perplexity. In contrast, only 38.1% of all reasoning steps in the full set exceed this threshold.

This stark discrepancy suggests that high confidence in SRM outputs is not indicative of correct or deep reasoning, but rather reflects a tendency to fall back on memorized patterns from training data. We present representative examples in Table 1 and Table 2, where seemingly confident steps lead to incorrect conclusions despite low token-level perplexity.

We interpret this overconfidence as a symptom of **cognitive overload**: when faced with challenging reasoning junctures, the SRM fails to engage in exploratory or reflective thinking and instead generates superficially fluent but semantically shallow continuations, effectively giving up by defaulting to familiar sequences. This behavior motivates our design of the Cognitive Offload Trigger in TrigReason, which detects such states and delegates to a more capable model before critical errors occur.

Table 1: Example of incorrect reasoning steps with corresponding perplexity ratios (ppl_ratio).

| Reasoning Step | ppl_ratio |
|---|---|
| AP = sqrt[(1100² + (100014)²)/507²] = sqrt[(1,210,000 + 1,400,000)/507²] = sqrt[(2,610,000)/507²] = sqrt[2,610,000]/507. | 0.955 |
| $$\frac{b^2}{1+m^2} \cdot \frac{1}{120} = 1$$ So: $$b^2 = 120(1+m^2)$$ Therefore, $b^2 = 120(1+m^2)$ and $a^2 = \frac{120(1+m^2)}{6-5m^2}$. | 0.916 |
| Simplify numerator and denominator: 50625÷25=2025, 22*325=7325. So, 2025/7325. | 0.931 |
| Okay, so from n=1 to n=20, the losing positions (L) are: 2, 5, 6, 10, 11, 15, 16, 20. | 0.872 |
| Let me denote the diagonals as vectors $\vec{d_1}$ and $\vec{d_2}$, which are perpendicular. So, if the rhombus is centered at the origin, then the vertices can be expressed as $\frac{\vec{d_1}}{2}$, $\frac{\vec{d_2}}{2}$, $-\frac{\vec{d_1}}{2}$, and $-\frac{\vec{d_2}}{2}$. | 0.863 |
| Finally, when her walking speed is $s + \frac{1}{2} = 3$ km/h, the time taken is: $$\frac{9}{3} + t = 3 + t$$ Since $t = 4 - 3 = 1$ hour, the total time is: $$3 + 1 = 4 \text{ hours}$$ | 0.962 |
| For the second scenario: - Walking speed: $s + 2$ km/h - Total time: 2 hours and 24 minutes = 144 + 24 = 168 minutes - Time in the coffee shop: $t$ minutes - Walking time: $168 - t$ minutes | 0.960 |

Table 2: Example of incorrect reasoning steps with corresponding perplexity ratios (ppl_ratio).

| Reasoning Step | ppl_ratio |
|---|---|
| Subtracting Equation 1 from Equation 2:

$$\frac{540}{s+2} - \frac{540}{s} = 144 - 240 = -96$$
$$540 \left( \frac{1}{s+2} - \frac{1}{s} \right) = -96$$
$$540 \left( \frac{-2}{s(s+2)} \right) = -96$$
$$\frac{1080}{s(s+2)} = -96$$ | 0.974 |
| $$V = (sqrt((a + b - c)(a + c - b)(b + c - a)))/(6 * sqrt(2))$$ | 0.952 |
| I can eliminate $t$ and find the value of $s$:
$$\frac{9}{s} - \frac{9}{s+2} = 1.6$$
Simplifying this, I get:
$$\frac{18}{s(s+2)} = 1.6$$
Solving for $s$, I find:
$$s = 5 \text{ km/h}$$ | 0.876 |
| Finally, when walking at $s + \frac{1}{2} = 3$ km/h, the time taken for the walk is:
$$\frac{9}{3} = 3 \text{ hours} = 180 \text{ minutes}$$ | 0.935 |
| n — Outcome (Winning or Losing for Alice) ——— 1 — Winning (Alice takes 1) 2 — Losing (Alice leaves Bob with 1) 3 — Winning (Alice takes 1, leaving 2 for Bob) 4 — Winning (Alice takes 4) 5 — Losing (Alice leaves Bob with 4 or 1) 6 — Losing (Alice leaves Bob with 5 or 2) 7 — Winning (Alice leaves Bob with 6 or 3) 8 — Winning (Alice leaves Bob with 7 or 4) 9 — Losing (Alice leaves Bob with 8 or 5) 10 — Losing (Alice leaves Bob with 9 or 6) 11 — Winning (Alice leaves Bob with 10 or 7) 12 — Winning (Alice leaves Bob with 11 or 8) 13 — Losing (Alice leaves Bob with 12 or 9) 14 — Losing (Alice leaves Bob with 13 or 10) 15 — Winning (Alice leaves Bob with 14 or 11) 16 — Winning (Alice leaves Bob with 15 or 12) 17 — Losing (Alice leaves Bob with 16 or 13) 18 — Losing (Alice leaves Bob with 17 or 14) 19 — Winning (Alice leaves Bob with 18 or 15) 20 — Winning (Alice leaves Bob with 19 or 16) 21 — Losing (Alice leaves Bob with 20 or 17) 22 — Losing (Alice leaves Bob with 21 or 18) 23 — Winning (Alice leaves Bob with 22 or 19) 24 — Winning (Alice leaves Bob with 23 or 20) 25 — Losing (Alice leaves Bob with 24 or 21) 26 — Losing (Alice leaves Bob with 25 or 22) 27 — Winning (Alice leaves Bob with 26 or 23) 28 — Winning (Alice leaves Bob with 27 or 24) 29 — Losing (Alice leaves Bob with 28 or 25) 30 — Losing (Alice leaves Bob with 29 or 26) 31 — Winning (Alice leaves Bob with 30 or 27) 32 — Winning (Alice leaves Bob with 31 or 3 | 0.929 |

## D  COMPLETE LIST OF HESITATION WORDS

To operationalize the linguistic markers of Recovery Inability Risk, we define a set of hesitation words and phrases that indicate uncertainty, self-doubt, or backtracking in reasoning trajectories. These patterns are used to detect when the SRM enters a state of semantic hesitation. We implement a case-insensitive regular expression matcher to identify such expressions in generated text. The full list of hesitation patterns is shown in Table 3.

Table 3: List of hesitation words and phrases.

| Word/Phrase | | |
|---|---|---|
| wait | hmm | debatable |
| maybe | perhaps | could be |
| might be | possibly | on the other hand |
| alternatively | another possibility | or perhaps |
| actually | now that I think about it | I think I made a mistake |
| let me reconsider | not sure | I'm not entirely sure |
| this might be wrong | I could be mistaken | unless I'm wrong |
| thinking | unsure | confused |

## E  THE MAIN ALGORITHM OF TRIGREASON

The main algorithm of TrigReason is shown in Algorithm 1.

---

**Algorithm 1** TrigReason

---

**Input:** Question $x$, small reasoning model $S$, large reasoning model $L$, priming steps $n$, overload threshold $\rho$, rectification steps $m$

1: Initialize: $y \leftarrow [\,]$, rectify_step $\leftarrow 0$, $t \leftarrow 0$
2: **while** not finished **do**
3:      $t \leftarrow t + 1$
4:      **if** $t < n$ **then**                              ▷ Strategic Priming Trigger
5:          $y_t \sim p_L(\cdot \mid y_{<t}, x)$
6:      **else**
7:          $\left(y_t^S, \text{finished}, \text{ppl\_ratio}_t\right) \leftarrow \text{GenerateStep}(S, x, y_{<t})$
8:          **if** rectify_step $> 0$ **or** ppl_ratio$_t > \rho$ **then**     ▷ Cognitive Offload or Recovery Trigger
9:              $y_t \sim p_L(\cdot \mid y_{<t}, x)$
10:              **if** ppl_ratio$_t <= \rho$ **then**
11:                  rectify_step $\leftarrow$ rectify_step $- 1$
12:              **end if**
13:          **else**
14:              $y_t \leftarrow y_t^S$                      ▷ Accept small model output
15:              **if** Detect_hesitation$(y_t, y_{t-1}, y_{t-2}))$ **then**
16:                  rectify_step $\leftarrow m$            ▷ Intervention Request Trigger fires
17:              **end if**
18:          **end if**
19:      **end if**
20:      Append $y_t$ to $y$
21: **end while**
22: **return** $y$

**Output:** Reasoning trajectory $y$, final answer

---

## F  PERFORMANCE UNDER VARYING TOKEN BUDGETS

We evaluate the effect of varying thinking token budgets (2K, 4K, 8K, 16K, 32K) on performance using AIME24. As shown in Figure 11, TrigReason consistently outperforms the SRM-only baseline

across all settings and matches the performance of both LRM-only and SpecReason, demonstrating its effectiveness and generalization under constrained reasoning resources.

However, as the budget increases, TrigReason and SpecReason exhibit a relative performance gap compared to LRM-only reasoning. This suggests that while collaborative reasoning is highly efficient at lower budgets, it may introduce slight suboptimality when targeting very peak accuracy in resource-abundant settings.

Figure 11: Accuracy comparison under different token budgets.

# G    THEORETICAL CHARACTERIZATION OF TRIGREASON RELIABILITY

## G.1    SETUP

Let a reasoning trajectory consist of $T$ steps, divided into:

- **Routine steps** ($T_{\text{rout}}$): Low-complexity inferences, forming the majority of steps.
- **Complex steps** ($T_{\text{comp}}$): High-risk steps involving multi-hop logic or ambiguity resolution. We assume $T_{\text{comp}} \sim \text{Poisson}(\lambda)$ with $\lambda \ll T$, so $\mathbb{E}[T_{\text{comp}}] = \lambda$ and $\mathbb{E}[T_{\text{rout}}] = T - \lambda$.

The **initial strategy** determines the overall path quality, modeled by a random variable $\gamma \in (0, 1]$, drawn from a distribution $\Gamma^M$ depending on the model $M$ used for strategy selection. We assume:

$$\mathbb{E}[\Gamma^{\text{LRM}}] > \mathbb{E}[\Gamma^{\text{SRM}}],$$

reflecting LRM's superior strategic coherence.

Per-step success probabilities (before strategy $\gamma$ scaling) are:

- Routine step, SRM: $1 - \epsilon_r$    ($\epsilon_r$ is small)
- Routine step, LRM: $1 - \delta_r$    ($\delta_r \approx \epsilon_r$)
- Complex step, SRM: $1 - \epsilon_c$    ($\epsilon_c \gg \epsilon_r$)
- Complex step, LRM: $1 - \delta_c$    ($\delta_c \ll \epsilon_c$)

## G.2    TRIGGER MECHANISM

TrigReason employs:

- **Strategic Priming Trigger**: Adopt the strategy of LRM $\gamma^{\text{LRM}}$ in the initial steps.
- **Cognitive Overload Trigger**: Fires on complex steps with probability $\alpha_{\text{comp}}$. Empirically, $\alpha_{\text{comp}}$ is close to 1 (refer to. Appendix C).
- **Intervention Request Trigger**: We assume reflection is only invoked if the reasoning step failed. Reflection succeeds with probability $\rho_l$ (LRM) or $\rho_s$ (SRM), with $\rho_l > \rho_s$. If successful, the error is corrected and reasoning continues; otherwise, the trajectory fails.

## G.3    PROPOSITION 1 (EXPECTED FINAL ERROR PROBABILITY OF TRIGREASON)

Under the above model, the expected final error probability of TrigReason is approximately:

$$\begin{aligned} P_{\text{fail}}^{\text{Trig}} \approx &(T - \lambda)\left(1 - \mathbb{E}[\gamma^{\text{LRM}}] + \mathbb{E}[\gamma^{\text{LRM}}]\epsilon_r\right) \\ &+ \lambda\left[\alpha_{\text{comp}}(1 - \mathbb{E}[\gamma^{\text{LRM}}]\delta_c) + (1 - \alpha_{\text{comp}})(1 - \rho_l)(1 - \mathbb{E}[\gamma^{\text{LRM}}]\epsilon_c)\right], \end{aligned}$$

where $\gamma^{\text{LRM}} \sim \Gamma^{\text{LRM}}$ is the path quality induced by LRM's initial strategy.

**Proof:**

We use the approximation $\mathbb{P}_{\text{success}} \approx \exp\left(-\sum_{t=1}^{T}(1 - p_t)\right)$, so for small cumulative error, $P_{\text{fail}} = 1 - \mathbb{P}_{\text{success}} \approx \mathbb{E}\left[\sum_{t=1}^{T}(1 - p_t)\right]$.

Routine steps: Each uses SRM, so success probability is $\mathbb{E}[\gamma^{\mathrm{LRM}}](1 - \epsilon_r)$. Expected error per step: $1 - \mathbb{E}[\gamma^{\mathrm{LRM}}](1 - \epsilon_r) \approx 1 - \mathbb{E}[\gamma^{\mathrm{LRM}}] + \mathbb{E}[\gamma^{\mathrm{LRM}}]\epsilon_r$. Total contribution: $(T - \lambda)(1 - \mathbb{E}[\gamma^{\mathrm{LRM}}] + \mathbb{E}[\gamma^{\mathrm{LRM}}]\epsilon_r)$.

Complex steps:

- With probability $\alpha_{\mathrm{comp}}$: Trigger fires, LRM generates the step. Error: $1 - \mathbb{E}[\gamma^{\mathrm{LRM}}]\delta_c$.

- With probability $1 - \alpha_{\mathrm{comp}}$: SRM generates the step. It fails with probability $1 - \mathbb{E}[\gamma^{\mathrm{LRM}}]\epsilon_c$. The next step triggers reflection, which succeeds with probability $\rho_l$, so the residual error after failed reflection is $(1 - \rho_l)(1 - \mathbb{E}[\gamma^{\mathrm{LRM}}]\epsilon_c)$. The expected error contribution is $(1 - \alpha_{\mathrm{comp}})(1 - \rho_l)(1 - \mathbb{E}[\gamma^{\mathrm{LRM}}]\epsilon_c)$.

Total expected error from complex steps:

$$\lambda \left[ \alpha_{\mathrm{comp}}(1 - \mathbb{E}[\gamma^{\mathrm{SRM}}]\delta_c) + (1 - \alpha_{\mathrm{comp}})(1 - \rho_l)(1 - \mathbb{E}[\gamma^{\mathrm{SRM}}]\epsilon_c) \right]$$

Summing both contributions yields the stated approximation.

## G.4 PROPOSITION 2 (EXPECTED COST OF TRIGREASON)

Let $c_s$ and $c_l$ be the cost of SRM and LRM steps, respectively, with $c_s \ll c_l$. The expected total cost of TrigReason is:

$$C^{\mathrm{Trig}} = Tc_s + (T_{stra} + \lambda)c_l.$$

**Proof:**

All $T$ steps are initially generated by SRM, incurring cost $Tc_s$. LRM is invoked in three cases:

- For **strategic priming steps**: $T_{stra}$ steps for adopting the strategy of LRM in the initial steps.

- For **triggered complex steps**: $\alpha_{\mathrm{comp}} \cdot T_{\mathrm{comp}}$, with expected count $\alpha_{\mathrm{comp}}\lambda$.

- For **reflection after errors**: $(1 - \alpha_{\mathrm{comp}}) \cdot T_{\mathrm{comp}}$, with expected count $(1 - \alpha_{\mathrm{comp}})\lambda$.

The total expected number of LRM calls is $T_{stra} + \lambda(\alpha_{\mathrm{comp}} + 1 - \alpha_{\mathrm{comp}}) = T_{stra} + \lambda$. Each call costs $c_l$, so the additional cost is $\lambda c_l$. The total expected cost is therefore $Tc_s + (T_{stra} + \lambda)c_l$.

## G.5 COMPARISON WITH LRM REASONING

Let $P_{\mathrm{fail}}^{\mathrm{LRM}}$ and $C^{\mathrm{LRM}}$ denote the failure probability and cost when LRM performs all reasoning steps and selects the initial strategy. Then:

$$P_{\mathrm{fail}}^{\mathrm{LRM}} \approx (T - \lambda)(1 - \mathbb{E}[\gamma^{\mathrm{LRM}}] + \mathbb{E}[\gamma^{\mathrm{LRM}}]\delta_r) + \lambda(1 - \mathbb{E}[\gamma^{\mathrm{LRM}}]\delta_c), \quad C^{\mathrm{LRM}} = Tc_l.$$

Since $\delta_c \ll \epsilon_c$, the dominant error terms in $P_{\mathrm{fail}}^{\mathrm{Trig}}$ are suppressed by either high trigger recall ($\alpha_{\mathrm{comp}}$) and strong reflection ($\rho_l$). Thus, $P_{\mathrm{fail}}^{\mathrm{Trig}}$ is close to $P_{\mathrm{fail}}^{\mathrm{LRM}}$, differing only in higher-order terms from routine steps.

However, because $\lambda + T_{stra} \ll T$ and $c_s \ll c_l$, we have:

$$C^{\mathrm{Trig}} = Tc_s + (T_{stra} + \lambda)c_l \ll Tc_l = C^{\mathrm{LRM}}.$$

Therefore, TrigReason achieves near-LRM reliability at a fraction of the computational cost, demonstrating the effectiveness of its targeted intervention design.

## G.6 COMPARISON WITH SRM REASONING

Let $P_{\mathrm{fail}}^{\mathrm{SRM}}$ and $C^{\mathrm{SRM}}$ denote the failure probability and cost when only the SRM is used for all steps, without any intervention. Then:

$$P_{\mathrm{fail}}^{\mathrm{SRM}} \approx (T - \lambda)\left(1 - \mathbb{E}[\gamma^{\mathrm{SRM}}] + \mathbb{E}[\gamma^{\mathrm{SRM}}]\epsilon_r\right) + \lambda(1 - \mathbb{E}[\gamma^{\mathrm{SRM}}]\epsilon_c), \quad C^{\mathrm{SRM}} = Tc_s.$$

Compared to TrigReason, its failure probability is significantly higher due to $\mathbb{E}[\Gamma^{\text{LRM}}] > \mathbb{E}[\Gamma^{\text{SRM}}]$, $\delta_c \ll \epsilon_c$ and $\rho_l > \rho_s$. Thus, we have:

$$P_{\text{fail}}^{\text{Trig}} \ll P_{\text{fail}}^{\text{SRM}}.$$

This demonstrates that TrigReason achieves a dramatic reliability improvement over SRM, by intelligently allocating LRM resources to high-risk steps.

## H  THE USE OF LARGE LANGUAGE MODELS

In this work, large language models are used exclusively to assist with language editing and clarification during the writing of this paper. All technical ideas, method design, analysis, and experimental work are conducted by human authors.

