# OpenReview forum: "TrigReason: Trigger-Based Collaboration between Small and Large Reasoning Models"
_ICLR.cc/2026/Conference — ICLR 2026 Conference Withdrawn Submission_

### Official Review · Reviewer_1WNG · 2025-10-27

**Soundness:** 2
**Presentation:** 2
**Contribution:** 2
**Rating:** 2
**Confidence:** 4

**Summary:**

LRMs suffer from high inference latency due to autoregressive token generation. Current acceleration methods using SRMs, such as SpecReason, rely on an inefficient polling-based design that invokes the LRM for verification at every step. This paper proposes TrigReason, a trigger-based collaborative framework that replaces continuous polling with selective intervention. TrigReason delegates most reasoning to the SRM and activates the LRM only when necessary: for initial strategic planning (Strategic Priming), upon detecting SRM overconfidence on difficult steps (Cognitive Offload), or when the SRM gets stuck (Intervention Request). This maintains LRM-level accuracy while significantly reducing latency and API costs in edge-cloud setups compared to SpecReason.

**Strengths:**

The paper provides a clear characterization of the limitations in prior work, specifically the polling-based SpecReason framework. It highlights the inefficiency of constant verification and presents experiments questioning the reliability of the LRM-as-a-Judge mechanism, noting that LRMs frequently reject their own reasoning steps.
The paper attempts to categorize the common failure modes of SRMs. It identifies three risk patterns: path divergence , cognitive overload , and recovery inability, which are then used to motivate the design of the proposed heuristic triggers.

The TrigReason framework is evaluated in an edge-cloud deployment scenario, a relevant setting for this type of collaborative inference. In this setting, the method is reported to reduce latency by 43.9% and API cost by 73.3% compared to the SpecReason baseline, addressing the overheads identified in the polling-based approach.

**Weaknesses:**

1. **Obvious writing error in the paper:** In Section 4.1 “Models”, the definitions of SRM and LRM are completely reversed. For example, the paper defines the 32B QwQ model as the SRM and the 1.5B DeepSeek model as the LRM. This directly contradicts other parts of the paper (e.g., Fig. 1 and Fig. 4) and represents a serious typographical mistake.

2. **Too many hyperparameters and manual configurations, questionable robustness:**
   The proposed method introduces several new hyperparameters, such as the *cognitive overload threshold*, *bootstrapping steps*, and *refinement steps*. The ablation study (Fig. 6a) shows that the optimal overload threshold varies across different model pairs. This suggests that the method may require careful hyperparameter tuning when applied to new SRM–LRM combinations, increasing the deployment cost and raising concerns about generalization and elegance.

   * **“Cognitive offloading” trigger (the trickiest part):** This trigger relies on a counterintuitive empirical assumption that when the SRM (the small model) experiences cognitive overload, it paradoxically becomes overconfident (i.e., exhibits unusually low perplexity). This is a very tricky observation rather than a general confidence estimation; it captures an abnormal behavior pattern specific to SRM’s failure mode. Moreover, the activation of this trigger depends on two finely tuned hyperparameters: the coverage threshold and perplexity threshold.
   * **“Intervention request” trigger:** This trigger essentially functions as a keyword-matching system. It scans for “hesitation words” such as *wait*, *hmm*, or *alternatively* in the text to determine whether the SRM is stuck. This is a clear heuristic rule that relies heavily on the predefined vocabulary (see Appendix D), thus lacking sufficient generalization capability.
   * **“Strategic initialization” trigger:** This is a hard-coded rule that forces LRM to take charge of the first n steps. Although the ablation study shows its effectiveness, this “forced start” design appears straightforward and lacks adaptivity.

3. **Dependence on heuristic rules:**
   The *intervention request trigger* depends on a predefined lexicon of hesitation words. Such word-based heuristic rules are not robust. If the SRM becomes logically stuck without using these specific words, the trigger may fail to activate. Overall, the mechanism appears overly *tricky* and insufficiently generalizable.

**Questions:**

Same as the weeknesses.

---

> ### Author Response · Authors · 2025-11-23
> **Rebuttal by Authors**
>
> Thank you for your detailed review and valuable feedback! Below, we address the comments you raised:
>
> **W1**: **Obvious writing error in the paper:** In Section 4.1 “Models”, the definitions of SRM and LRM are completely reversed. For example, the paper defines the 32B QwQ model as the SRM and the 1.5B DeepSeek model as the LRM. This directly contradicts other parts of the paper (e.g., Fig. 1 and Fig. 4) and represents a serious typographical mistake.
>
> **R**: We sincerely thank the reviewer for identifying this critical typo. This was an unfortunate typographical error, and we have corrected it in the latest version of the manuscript.
>
> **W2**: **Too many hyperparameters and manual configurations, questionable robustness:** The proposed method introduces several new hyperparameters, such as the cognitive overload threshold, bootstrapping steps, and refinement steps. The ablation study (Fig. 6a) shows that the optimal overload threshold varies across different model pairs. This suggests that the method may require careful hyperparameter tuning when applied to new SRM–LRM combinations, increasing the deployment cost and raising concerns about generalization and elegance.
>
> **R**: We would like to address your concerns through the following points:
>
> **(1) On hyperparameter tuning:**   The goal of tuning TrigReason’s parameters is to find optimal trade-off between efficiency and accuracy. However, exact optimality is not required in practice, as moderate redundancy is acceptable and still yields significant gains. For example, in our ablation on the Cognitive Overload Threshold (ρ), performance already matches full LRM when ρ ≤ 0.85. Even with a more conservative setting like ρ = 0.75, nearly half of all tokens are generated by the SRM, delivering substantial speedup with no accuracy drop.
>
> **(2) On cross-model transferability:**  The main factor affecting tuning ρ is the inherent perplexity of the SRM, which varies across models due to differences in model architecture and training process. For instance, we conduct a micro-benchmark on WikiText-v2 dataset, DeepSeek-R1-1.5B has an average perplexity of 6.56, while Qwen3-0.6B is at 4.22. This naturally leads to different sensitivities to the Cognitive Overload Trigger, requiring model-specific calibration of ρ, but only once per SRM.
>
> **(3) On cross-domain generalization:**  When the SRM–LRM pair is fixed, the same parameter configuration generalizes well across reasoning domains. As shown in our experiments on Big Bench Hard (BBH) (complex logical reasoning) and ARC-Challenge (commonsense reasoning) using DeepSeek-R1-1.5B + QwQ-32B, we kept all parameters identical to those used on AIME, except reducing Priming Steps from 20 to 5 (due to lower task complexity). Results:
>
> | Metric     | SRM       | LRM       | TrigReason |
> |----|----|----|----|
> | Accuracy   |    |   |   |
> | BBH        | 42.2%   | 67.5%     | **68.7%**  |
> | ARC        | 59.3%   | 95.7%     | **94.8%**  |
> | Latency       |       |         |         |
> | BBH        | 34.4 min  | 178.2 min | **125.7 min** |
> | ARC        | 7.0 s     | 22.9 s    | **15.6 s**  |
>
> TrigReason achieves near-LRM or even better accuracy while having lots of reasoning steps generated by SRM, demonstrating both parameter robustness and domain generality beyond math/QA tasks.
>
> **W2.1**: “Cognitive offloading” trigger (the trickiest part): This trigger relies on a counterintuitive empirical assumption that when the SRM (the small model) experiences cognitive overload, it paradoxically becomes overconfident (i.e., exhibits unusually low perplexity). This is a very tricky observation rather than a general confidence estimation; it captures an abnormal behavior pattern specific to SRM’s failure mode. Moreover, the activation of this trigger depends on two finely tuned hyperparameters: the coverage threshold and perplexity threshold.
>
> **R**: We thank the reviewer for this thoughtful comment, but we respectfully disagree with the characterization of the cognitive offloading trigger as a *tricky or counterintuitive empirical assumption*. We address this concern in three parts:
>
> **(1) On the role of perplexity**:  The interpretation of perplexity (or token entropy) in reasoning models remains an open research question. While some works show that reducing entropy can improve reasoning performance [1, 2], others advocate maintaining high entropy to encourage uncertainty during post-training [3,4]. In addition, there are also studies find that LLMs express overconfidence by attempting to rationalize the incorrect outputs [5].
>
> [1] The unreasonable effectiveness of entropy minimization in llm reasoning.
>
> [2] Confidence is all you need: Few-shot rl fine-tuning of language models.
>
> [3] The entropy mechanism of reinforcement learning for reasoning language models.
>
> [4] Reasoning with exploration: An entropy perspective.
>
> [5] Learn to be Honest: Mitigate LLMs' Overconfidence for Improving Hallucination Detection with Self-Hesitation Activation.

---

> ### Author Response · Authors · 2025-11-23
> **Rebuttal by Authors**
>
> **(2) On robustness and detection accuracy** : As shown in Appendix C, among 93 trajectories with identifiable errors, 88 (94.6%) contain at least one step where over 85% of tokens have low perplexity. In contrast, only 38.1% of all reasoning steps exceed this threshold across the full dataset. This indicates that although the Cognitive Overload Trigger is not a perfect error detector, which have both False negatives (errors missed by the trigger, but they are rare in statistics) and False positives (trigger fires without actual error, more common but acceptable given the significant efficiency gains from avoiding full LRM reasoning). In practice, our empirical analysis shows that low-perplexity steps are strongly correlated with reasoning breakdown, making them effective proxies for intervention.
>
> **(3) On hyperparameters sensitivity and generalization:**  As noted in our response to **W2**, the goal of tuning TrigReason’s parameters is to find optimal trade-off between efficiency and accuracy, not aimed at maximizing specific benchmark performance. If peak efficiency is not required, only coarse tuning is necessary and precise fine-tuning is optional.
>
> Moerover, the coverage threshold ($\rho$) and perplexity threshold ($\tau$)  need adjustment primarily due to inherent differences across SRMs (their baseline perplexity levels), as explained in our earlier response.
>
> Finally, to further address concerns about sensitivity to these thresholds, we conducted additional ablation studies varying $\rho$ and $\tau$, measuring both trigger activation rates and final accuracy.
>
> | $\tau$ | Cognitive Offload (%) | Strategic Priming (%) | Intervention Request (%) | Total Trigger (%) | AIME24 ACC |
> |----|----|-----|-------|---------|------------|
> | 1.02   | 22.22%     | 7.89%     | 4.82%       | 34.93%    | 31.70%     |
> | 1.05   | 25.95%   | 8.31%   | 4.73%      | 38.99%     | 29.6%      |
> | 1.10   | 30.08%     | 8.42%     | 5.28%      | 43.78%     | 30.00%     |
>
> | Parameter (p-n-m) | Cognitive Offload (%) | Strategic Priming (%) | Intervention Request (%) | Total Trigger (%) | AIME24 ACC | LRM Acc = 29.2 |
> |---------|----|-----|------|---|-----|-------|
> | 0.75-20-1   | 38.50% | 8.06%   | 4.36%    | 50.92%   | 30.0  |     |
> | 0.85-20-1| 26.95%  | 8.31%  | 4.73%    | 39.99%     | 29.6    |     |
> | 0.95-20-1  | 11.49%    | 8.64%    | 4.95%    | 25.08%    | 26.25   |     |
>
> Results show that $\tau$ and $\rho$ jointly control the frequency of the Cognitive Offload trigger, but as long as the trigger activates on a sufficient fraction of reasoning steps, enabling it to capture most SRM failures, accuracy remains stable. The main effect of parameter variation is on efficiency.
>
> **W2.2**: “Intervention request” trigger: This trigger essentially functions as a keyword-matching system. It scans for “hesitation words” such as wait, hmm, or alternatively in the text to determine whether the SRM is stuck. This is a clear heuristic rule that relies heavily on the predefined vocabulary (see Appendix D), thus lacking sufficient generalization capability.
>
> **W3**: Dependence on heuristic rules: The intervention request trigger depends on a predefined lexicon of hesitation words. Such word-based heuristic rules are not robust. If the SRM becomes logically stuck without using these specific words, the trigger may fail to activate. Overall, the mechanism appears overly tricky and insufficiently generalizable.
>
> **R**: Hesitation-based triggers are designed based on efficiency considerations. Although they are effective in practice, they still raise concerns about generalization. Therefore, we design a hesitancy sentiment classifier to replace the detection of hesitation words. Specifically, we first collect 12k labeled reasoning steps from AIME, GPQA, and BBH, annotated using Qwen3-32B. And then, We fine-tune BERT-Tiny (a lightweight model with only 2 transformer layers) for binary hesitation classification, using a linear classifier on top of the [CLS] token representation, following standard BERT-based sequence classification. Finally, We integrate this trained classifier into TrigReason’s pipeline to replace the original hesitation words based detector. The benchmark results are shown in the table below:
>
> | Parameter (ρ-n-m) | Cognitive Offload (%) | Strategic Priming (%) | Intervention Request (%) | Total Trigger (%) | AIME24 ACC | AIME25 ACC |
> |----|------|-----|----------|-------|------------|-----|
> | 0.85-20-1 (original)       | 25.95%  | 8.31% | 4.73%     | 38.99%     | 29.6  | 29.58 |
> | 0.85-20-1 (with classifier) | 25.22%| 8.47%  | 6.21%       | 39.90%    | 29.6 | 30.00  |
>
> It can be seen that the learned detector achieves higher activation frequency of Intervention Request, indicating broader coverage.
> It also leads to a slight improvement in accuracy on AIME25 dataset. We will make this learned trigger option configurable in TrigReason, allowing users to choose between heuristic and learned variants based on their needs.

---

> ### Author Response · Authors · 2025-11-23
> **Rebuttal by Authors**
>
> **W2.3**: “Strategic initialization” trigger: This is a hard-coded rule that forces LRM to take charge of the first n steps. Although the ablation study shows its effectiveness, this “forced start” design appears straightforward and lacks adaptivity.
>
> **R**: While the “Strategic Initialization” trigger is indeed a simple rule, it serves a distinct purpose from the Cognitive Offload trigger: it prevents macro-level planning failures caused by the SRM’s limited ability to chart a correct high-level solution path at the outset. As our ablation study shows, this design is highly effective despite its simplicity.
>
> In fact, this trigger can be made adaptive when an estimate of total reasoning length is available. For example, using a pre-run on AIME24, we obtained the average number of steps per problem and set \( n \) to 5% of the total steps. With this adaptive setting, TrigReason achieves comparable accuracy to the fixed $n=20$ configuration, while reducing the Strategic Priming trigger activation frequency by 3%, further improving efficiency. However, in most real-world scenarios, the required reasoning length is not known in advance, making such adaptation impractical. In these cases, a fixed “forced start” provides a reliable, low-overhead safeguard, making it a practical and justified design choice for general deployment.
>
> We hope this response has adequately addressed your concerns. If you have any further queries or require additional information, please feel free to let us know. We look forward to the opportunity for additional discussion during the review process.

---

> > ### Comment · Reviewer_1WNG · 2025-11-26
> > **Response**
> >
> > Thank you for the detailed and thoughtful response. I appreciate the time and effort you invested, especially given the short timeframe, to provide additional analyses and results. Some of your clarifications and proposed solutions do address part of my concerns, and I will raise my score to 4.
> > There is a Chinese idiom that says “great skill appears clumsy,” (da qiao bu gong) and as I mentioned in my initial review, the core issue remains that the method introduces many fine-grained designs and heuristic rules. Many of these components may work on the chosen benchmarks, but their robustness is uncertain and they may easily fail in real-world deployment scenarios or on new LLMs. I think that, your work may indeed align well with the stylistic preferences of conferences such as KDD? Thank you again for your time and effort.

---

### Official Review · Reviewer_xCXr · 2025-10-27

**Soundness:** 2
**Presentation:** 2
**Contribution:** 2
**Rating:** 4
**Confidence:** 3

**Summary:**

This paper introduces TrigReason, a trigger-based collaborative reasoning framework designed to improve the efficiency of speculative reasoning between Small Reasoning Models (SRMs) and Large Reasoning Models (LRMs). Current approaches rely on a polling-based mechanism where the LRM verifies each reasoning step generated by the SRM. This frequent verification is computationally expensive and unreliable because LRMs often make subjective or inconsistent judgments about intermediate reasoning steps. To overcome this, TrigReason replaces continuous polling with selective, event-triggered intervention. It identifies and addresses three key reasoning failure modes of SRMs, including (1) Path Divergence, (2) Cognitive Overload, and (3) Recovery Inability, and introduces corresponding triggers. Empirical results across three benchmarks show that TrigReason maintains or slightly surpasses LRM-level accuracy while reducing latency and API cost under edge–cloud settings, compared to SpecReason.

**Strengths:**

1. The paper identifies a concrete bottleneck, i.e., polling inefficiency in speculative reasoning, and provides convincing empirical evidence that LRM judgments are inconsistent and unreliable, even when judging their own reasoning trajectories.
2. The paper proposes an event-driven approach grounded in behavioral analysis of SRM reasoning failures. The introduction of token-level perplexity monitoring and linguistic hesitation detection represents a good diagnostic approach for hybrid reasoning systems.
3. Extensive experiments across multiple reasoning datasets and SRM–LRM pairs, together with edge–cloud deployment studies, demonstrate both generality and practical efficiency.

**Weaknesses:**

1. While empirically well-supported, the triggers lack a formal theoretical foundation. The thresholds (e.g., perplexity < 1.05) and trigger rules are heuristic, raising questions about generalizability to unseen models or tasks.
2. The detection mechanisms (e.g., fixed hesitation word list, perplexity thresholds) depend on manually designed signals rather than learned or adaptive ones, which might limit scalability to broader reasoning domains beyond math and logic.
3. The benchmarks focus heavily on symbolic/mathematical reasoning (AIME, GPQA). It remains unclear how well TrigReason extends to open-ended reasoning, commonsense tasks.
4. There are many presentation issues in the paper, such as incomplete sentence in Line 189, the direct reference to the figures in Appendix (this is potentially violating the page limit policy).

**Questions:**

1. How sensitive are the trigger parameters across different model pairs or domains? Is it possible to set them more automatically?
2. Have you tested TrigReason on tasks involving non-symbolic reasoning (e.g., commonsense QA)?
3. How do the triggers behave when SRM outputs are noisy or contain linguistic variations (e.g., “hmm...” vs “let’s rethink this”)?
4. In edge–cloud setups, how does TrigReason handle asynchronous LRM availability or network delays? Could trigger latency itself introduce new bottlenecks?

**Details Of Ethics Concerns:**

In the main body of the submission, the authors directly refer to the figures in the Appendix, instead of the sections in the Appendix. This potentially violates the page limit policy (9 pages at most). In other venues, such unprofessional presentations will result in desk rejection.

---

> ### Author Response · Authors · 2025-11-23
> **Rebuttal by Authors**
>
> Thank you for your detailed review and valuable feedback! Below, we address the comments you raised:
>
> **W1**: While empirically well-supported, the triggers lack a formal theoretical foundation. The thresholds (e.g., perplexity < 1.05) and trigger rules are heuristic, raising questions about generalizability to unseen models or tasks.
>
> **Q1**: How sensitive are the trigger parameters across different model pairs or domains? Is it possible to set them more automatically?
>
> **R**: We would like to clarify these important questions through the following points:
>
> **(1) On hyperparameter tuning:**   The goal of tuning TrigReason’s parameters is to find optimal trade-off between efficiency and accuracy. However, exact optimality is not required in practice, as moderate redundancy is acceptable and still yields significant gains. For example, in our ablation on the Cognitive Overload Threshold (ρ), performance already matches full LRM when ρ ≤ 0.85. Even with a more conservative setting like ρ = 0.75, nearly half of all tokens are generated by the SRM, delivering substantial speedup with no accuracy drop.
>
> **(2) On cross-model transferability:**  The main factor affecting tuning ρ is the inherent perplexity of the SRM, which varies across models due to differences in model architecture and training process. For instance, we conduct a micro-benchmark on WikiText-v2 dataset, DeepSeek-R1-1.5B has an average perplexity of 6.56, while Qwen3-0.6B is at 4.22. This naturally leads to different sensitivities to the Cognitive Overload Trigger, requiring model-specific calibration of ρ, but only once per SRM.
>
> **(3) On cross-domain generalization:**  When the SRM–LRM pair is fixed, the same parameter configuration generalizes well across reasoning domains. As shown in our experiments on Big Bench Hard (BBH) (complex logical reasoning) and ARC-Challenge (commonsense reasoning) using DeepSeek-R1-1.5B + QwQ-32B, we kept all parameters identical to those used on AIME, except reducing Priming Steps from 20 to 5 (due to lower task complexity). Results:
>
> | Metric     | SRM       | LRM       | TrigReason |
> |------------|-----------|-----------|------------|
> | Accuracy   |           |           |            |
> | BBH        | 42.2%     | 67.5%     | **68.7%**  |
> | ARC        | 59.3%     | 95.7%     | **94.8%**  |
> | Latency       |           |           |            |
> | BBH        | 34.4 min  | 178.2 min | **125.7 min** |
> | ARC        | 7.0 s     | 22.9 s    | **15.6 s**  |
>
> TrigReason achieves near-LRM or even better accuracy while having lots of reasoning steps generated by SRM, demonstrating both parameter robustness and domain generality beyond math/QA tasks.
>
> Regarding setting parameters more automatically, hyperparameters in TrigReason can be tuned once via a grid search on the SRM using a representative validation task. As shown in our experiments (e.g., BBH, ARC), these settings generalize well across different reasoning domains and LRM partners, as long as the SRM remains unchanged. Thus, re-tuning is only needed when the SRM is replaced, making the process efficient and scalable in real-world deployment.
>
> **W3**: The benchmarks focus heavily on symbolic/mathematical reasoning (AIME, GPQA). It remains unclear how well TrigReason extends to open-ended reasoning, commonsense tasks.
>
> **Q2**: Have you tested TrigReason on tasks involving non-symbolic reasoning (e.g., commonsense QA)?
>
> **R**: As shown in the table above, we conduct experiments on two non-symbolic reasoning benchmarks:
> - **Big Bench Hard (BBH)**: a complex logical reasoning benchmark
> - **AI2 Reasoning Challenge (ARC-Challenge)**: a commonsense reasoning benchmark
>
> The results demonstrate the effectiveness in diverse reasoning domains of TrigReason.
>
> **W2**: The detection mechanisms (e.g., fixed hesitation word list, perplexity thresholds) depend on manually designed signals rather than learned or adaptive ones, which might limit scalability to broader reasoning domains beyond math and logic.
>
> **Q3**: How do the triggers behave when SRM outputs are noisy or contain linguistic variations (e.g., “hmm...” vs “let’s rethink this”)?
>
> **R**:  We design a hesitancy sentiment classifier to replace the detection of hesitation words for generalization. Specifically, we first collect 12k labeled reasoning steps from AIME, GPQA, and BBH, annotated using Qwen3-32B. And then, We fine-tune BERT-Tiny (a lightweight model with only 2 transformer layers) for binary hesitation classification, using a linear classifier on top of the [CLS] token representation, following standard BERT-based sequence classification. Finally, We integrate this trained classifier into TrigReason’s pipeline to replace the original hesitation words based detector. The benchmark results are shown in the table below:

---

> ### Author Response · Authors · 2025-11-23
> **Rebuttal by Authors**
>
> | Parameter (ρ-n-m) | Cognitive Offload (%) | Strategic Priming (%) | Intervention Request (%) | Total Trigger (%) | AIME24 ACC | AIME25 ACC |
> |-------------------|------------------------|------------------------|----------------------------|--------------------|------------|------------|
> | 0.85-20-1 (original)       | 25.95%                 | 8.31%                  | 4.73%                      | 38.99%             | 29.6       | 29.58      |
> | 0.85-20-1 (with classifier) | 25.22%           | 8.47%                  | 6.21%                      | 39.90%             | 29.6       | 30.00  |
>
> It can be seen that the learned detector achieves higher activation frequency of Intervention Request, indicating broader coverage.
> It also leads to a slight improvement in accuracy on AIME25 dataset. We will make this learned trigger option configurable in TrigReason, allowing users to choose between heuristic and learned variants based on their needs.
>
> **W4**: There are many presentation issues in the paper, such as incomplete sentence in Line 189, the direct reference to the figures in Appendix (this is potentially violating the page limit policy).
>
> **R**: We sincerely thank the reviewer for pointing out these issues. We have corrected the incomplete sentence on Line 189 and revised all appendix figure references in the updated manuscript.
>
> **Q4**: In edge–cloud setups, how does TrigReason handle asynchronous LRM availability or network delays? Could trigger latency itself introduce new bottlenecks?
>
> **R**: We observe that in certain edge–cloud scenarios, network congestion or remote API queuing (due to server-side load) can introduce additional latency for LRM inference steps in the collaboration. However, such delays are outside the control of TrigReason, they depend entirely on the cloud model’s availability and scheduling conditions.
>
> However, the key improvement of TrigReason over prior collaborative frameworks (e.g., SpecReason) lies in its trigger-based selective delegation, which significantly reduces the number of calls to LRM. For instance, as reported in the manuscript, under the edge–cloud setting, TrigReason reduces API usage to only 27% of that required by SpecReason. Therefore, while TrigReason cannot eliminate external network or service-side delays, it minimizes the number of LRM invocations, thereby introducing less exposure to such external overheads compared to existing methods.
>
> We hope this response has adequately addressed your concerns. If you have any further queries or require additional information, please feel free to let us know. We look forward to the opportunity for additional discussion during the review process.

---

> > ### Comment · Reviewer_xCXr · 2025-11-26
> >
> > Thanks for the responses. After reading other reviewers' comments, I decide to maintain my original rating. Although the authors addressed several presentation issues in the revision, the current version of the paper is still not ready for publication.

---

### Official Review · Reviewer_EARS · 2025-10-29

**Soundness:** 3
**Presentation:** 3
**Contribution:** 2
**Rating:** 6
**Confidence:** 3

**Summary:**

This paper introduces TrigReason, a framework aiming to enhance the efficiency of collaborative reasoning between a Small Reasoning Model (SRM) and a Large Reasoning Model (LRM). It critiques the polling-based approach of prior work like SpecReason, where the LRM verifies every SRM step, citing inefficiency and unreliability of LRM judgments. TrigReason identifies three common SRM reasoning risks: path divergence (poor initial strategy), cognitive overload (failure on complex steps, often signaled by overconfidence/low perplexity), and recovery inability (getting stuck, signaled by hesitation markers) . Instead of continuous polling, TrigReason uses selective LRM intervention triggered by specific events: (1) Strategic Priming (LRM provides initial steps), (2) Cognitive Offload (LRM intervenes on overconfident SRM steps), and (3) Intervention Request (LRM intervenes upon detecting SRM hesitation loops). Experiments on math (AIME24/25) and QA (GPQA) benchmarks show TrigReason maintains accuracy comparable to LRM-only and SpecReason, while significantly improving efficiency: increasing SRM token utilization (1.7x-4.8x vs SpecReason), and reducing latency (by 43.9%) and API cost (by 73.3%) in simulated edge-cloud settings. Ablations support the trigger designs, and a theoretical reliability characterization is included.

**Strengths:**

**Well-Motivated Problem:** Addresses the important practical challenge of reducing LRM inference latency and cost for complex reasoning tasks while maintaining accuracy.

**Principled Approach:** The trigger-based design is grounded in a systematic analysis of SRM failure modes, making the interventions targeted and interpretable.

**Significant Efficiency Gains:** The empirical results convincingly demonstrate substantial improvements in latency, cost, and SRM utilization compared to the SpecReason baseline, without compromising accuracy.

**Thorough Evaluation:** The framework is validated across multiple difficult benchmarks (AIME, GPQA) , various SRM-LRM pairs , including a practical edge-cloud simulation , and supported by ablation studies.


**Clear Improvement over Baseline:** Directly tackles and demonstrably overcomes key weaknesses (inefficiency, unreliability) identified in the prior state-of-the-art (SpecReason).

**Weaknesses:**

**Hyperparameter Tuning:** The framework introduces several hyperparameters (priming steps $n$, perplexity threshold $\tau$, coverage threshold $\rho$, hesitation count $k$, rectification steps $m$). While ablations show sensitivity, the optimal values appear model/task-dependent. Guidance on tuning these efficiently for new setups is limited.

**Domain Generality:** The evaluation is primarily focused on mathematical reasoning and QA tasks. While these are important domains, the universality of the identified failure modes and the effectiveness of the triggers in other reasoning domains (e.g., planning, creative generation, coding) remains to be demonstrated.

**Reliability of Trigger Signals:** The paper demonstrates correlations, but the causal link and robustness need consideration. Can SRMs fail without showing high confidence or hesitation? Can these signals appear spuriously? The reliance on a fixed list of hesitation words (Appendix D) might also be brittle.

**Questions:**

1. How robust are the optimal hyperparameter settings found in the ablations (e.g., $\rho=0.85$ or $0.75$) across different SRM-LRM pairs, model sizes, or reasoning domains beyond math/QA? What is the recommended procedure for tuning these parameters in practice?

2. Regarding the Cognitive Offload trigger: How sensitive is performance to the perplexity threshold $\tau$? Are there failure cases where SRMs err without exhibiting abnormally low perplexity, or vice-versa? Were alternative confidence metrics considered?

3. Regarding the Intervention Request trigger: How was the list of hesitation words (Appendix D) derived? How well does this trigger generalize across different model families or languages? Could models enter unproductive loops without using these specific phrases?

4. How are discrete "reasoning steps" defined for calculating the perplexity ratio $r_s$ and tracking consecutive hesitations? Is segmentation based on punctuation, newlines, or another method, and how might this affect trigger activation?

5. Appendix F suggests TrigReason might lag behind LRM-only performance at very high token budgets (32K). Can you quantify this gap and explain why trigger-based collaboration might become suboptimal in resource-abundant settings?

---

> ### Author Response · Authors · 2025-11-23
> **Rebuttal by Authors**
>
> Thank you for your detailed review and valuable feedback! Below, we address the comments you raised:
>
> **W1**: **Hyperparameter Tuning**: The framework introduces several hyperparameters (priming steps $n$, perplexity threshold $\tau$, coverage threshold $\rho$, hesitation count $k$, rectification steps $m$). While ablations show sensitivity, the optimal values appear model/task-dependent. Guidance on tuning these efficiently for new setups is limited.
>
> **W2**: **Domain Generality**: The evaluation is primarily focused on mathematical reasoning and QA tasks. While these are important domains, the universality of the identified failure modes and the effectiveness of the triggers in other reasoning domains (e.g., planning, creative generation, coding) remains to be demonstrated.
>
> **Q1**: How robust are the optimal hyperparameter settings found in the ablations (e.g., $\rho = 0.85$ or $0.75$) across different SRM-LRM pairs, model sizes, or reasoning domains beyond math/QA? What is the recommended procedure for tuning these parameters in practice?
>
> **R**: We would like to clarify these important questions about hyperparameter robustness and domain generality through the following points:
>
> **(1) On hyperparameter tuning:**   The goal of tuning TrigReason’s parameters is to find optimal trade-off between efficiency and accuracy. However, exact optimality is not required in practice, as moderate redundancy is acceptable and still yields significant gains. For example, in our ablation on the Cognitive Overload Threshold (ρ), performance already matches full LRM when ρ ≤ 0.85. Even with a more conservative setting like ρ = 0.75, nearly half of all tokens are generated by the SRM, delivering substantial speedup with no accuracy drop.
>
> **(2) On cross-model transferability:**  The main factor affecting tuning ρ is the inherent perplexity of the SRM, which varies across models due to differences in model architecture and training process. For instance, we conduct a micro-benchmark on WikiText-v2 dataset, DeepSeek-R1-1.5B has an average perplexity of 6.56, while Qwen3-0.6B is at 4.22. This naturally leads to different sensitivities to the Cognitive Overload Trigger, requiring model-specific calibration of ρ, but only once per SRM.
>
> **(3) On cross-domain generalization:**  When the SRM–LRM pair is fixed, the same parameter configuration generalizes well across reasoning domains. As shown in our experiments on Big Bench Hard (BBH) (complex logical reasoning) and ARC-Challenge (commonsense reasoning) using DeepSeek-R1-1.5B + QwQ-32B, we kept all parameters identical to those used on AIME, except reducing Priming Steps from 20 to 5 (due to lower task complexity). Results:
>
> | Metric     | SRM       | LRM       | TrigReason |
> |------------|-----------|-----------|------------|
> | Accuracy   |           |           |            |
> | BBH        | 42.2%     | 67.5%     | **68.7%**  |
> | ARC        | 59.3%     | 95.7%     | **94.8%**  |
> | Latency       |           |           |            |
> | BBH        | 34.4 min  | 178.2 min | **125.7 min** |
> | ARC        | 7.0 s     | 22.9 s    | **15.6 s**  |
>
> TrigReason achieves near-LRM or even better accuracy while having lots of reasoning steps generated by SRM, demonstrating both parameter robustness and domain generality beyond math/QA tasks.
>
> **W3**: **Reliability of Trigger Signals**: The paper demonstrates correlations, but the causal link and robustness need consideration. Can SRMs fail without showing high confidence or hesitation? Can these signals appear spuriously? The reliance on a fixed list of hesitation words (Appendix D) might also be brittle.
>
> **Q2**: Regarding the Cognitive Offload trigger: How sensitive is performance to the perplexity threshold $\tau$? Are there failure cases where SRMs err without exhibiting abnormally low perplexity, or vice-versa? Were alternative confidence metrics considered?
>
> **R**: As described in Appendix C, among the 93 trajectories with identifiable errors, 88 (94.6%) contain steps where over 85% of tokens are low-perplexity. Incontrast, only 38.1% of all reasoning steps in the full set exceed this threshold. This indicates that the Cognitive Overload Trigger is not a perfect error detector, which have both False negatives (errors missed by the trigger, though they are rare in statistics) and False positives (trigger fires without actual error, more common but acceptable given the significant efficiency gains from avoiding full LRM reasoning). We also note that SRMs can fail without showing low perplexity, and conversely, low perplexity does not always imply correctness, which aligns with known limitations of perplexity as a confidence metric. However, in practice, our empirical analysis shows that low-perplexity steps are strongly correlated with reasoning breakdown, making them effective proxies for intervention.

---

> ### Author Response · Authors · 2025-11-23
> **Rebuttal by Authors**
>
> Regarding perplexity threshold sensitivity, we further conduct ablation studies on $\tau$, keeping $\rho = 0.85$ fixed. Results are shown in the table below:
>
> | $\tau$ | Cognitive Offload (%) | Strategic Priming (%) | Intervention Request (%) | Total Trigger (%) | AIME24 ACC |
> |--------|------------------------|------------------------|----------------------------|--------------------|------------|
> | 1.02   | 22.22%                 | 7.89%                  | 4.82%                      | 34.93%             | 31.70%     |
> | 1.05   | 25.95%                 | 8.31%                  | 4.73%                      | 38.99%             | 29.6%      |
> | 1.10   | 30.08%                 | 8.42%                  | 5.28%                      | 43.78%             | 30.00%     |
>
> Here, (%) denotes the proportion of reasoning steps that trigger LRM. As shown, $\tau$ and $\rho$ jointly control the activation frequency of the Cognitive Overload trigger. Notably, reducing $\tau$ slightly improves detection precision: it lowers the trigger rate (enhancing efficiency) while maintaining high accuracy, suggesting that a tighter perplexity threshold better isolates actual SLM failures.
>
> **W3**: The reliance on a fixed list of hesitation words (Appendix D) might also be brittle.
>
> **Q3**: Regarding the Intervention Request trigger: How was the list of hesitation words (Appendix D) derived? How well does this trigger generalize across different model families or languages? Could models enter unproductive loops without using these specific phrases?
>
> **R**: The hesitation word list is derived through a two-stage analysis of 960 reasoning traces generated by DeepSeek-R1-1.5B and Qwen3-0.6B on the AIME24 dataset. First, we identified reasoning steps exhibiting hesitation or stagnation by prompting LRM like QwQ-32B. And then we performed n-gram frequency analysis and compiled a candidate set of high-frequency patterns. Finally, we manually curated this candidate set to retain only those expressions that consistently convey hesitation in context.
>
> Hesitation-based triggers are designed based on efficiency considerations. Although they are effective in practice, they still raise concerns about generalization. Therefore, we design a hesitancy sentiment classifier to replace the detection of hesitation words. Specifically, we first collect 12k labeled reasoning steps from AIME, GPQA, and BBH, annotated using Qwen3-32B. And then, We fine-tune BERT-Tiny (a lightweight model with only 2 transformer layers) for binary hesitation classification, using a linear classifier on top of the [CLS] token representation, following standard BERT-based sequence classification. Finally, We integrate this trained classifier into TrigReason’s pipeline to replace the original hesitation words based detector. The benchmark results are shown in the table below:
>
> | Parameter (ρ-n-m) | Cognitive Offload (%) | Strategic Priming (%) | Intervention Request (%) | Total Trigger (%) | AIME24 ACC | AIME25 ACC |
> |-------------------|------------------------|------------------------|----------------------------|--------------------|------------|------------|
> | 0.85-20-1 (original)       | 25.95%                 | 8.31%                  | 4.73%                      | 38.99%             | 29.6       | 29.58      |
> | 0.85-20-1 (with classifier) | 25.22%           | 8.47%                  | 6.21%                      | 39.90%             | 29.6       | 30.00  |
>
> It can be seen that the learned detector achieves higher activation frequency of Intervention Request, indicating broader coverage.
> It also leads to a slight improvement in accuracy on AIME25 dataset. We will make this learned trigger option configurable in TrigReason, allowing users to choose between heuristic and learned variants based on their needs.
>
>
> **Q4**: How are discrete "reasoning steps" defined for calculating the perplexity ratio $r_s$ and tracking consecutive hesitations? Is segmentation based on punctuation, newlines, or another method, and how might this affect trigger activation?
>
> **R**: In our framework, discrete reasoning steps are explicitly induced via prompt engineering: we instruct the model to “separate logical reasoning steps with two newline characters (\n\n)”. This design choice provides a clear, consistent segmentation signal that aligns with human-like step-by-step reasoning, enabling reliable computation of per-step perplexity ratios and detection of consecutive hesitation patterns.
>
> We acknowledge that non-compliant outputs (e.g., merging multiple logical steps into one block) could theoretically reduce trigger sensitivity. However, in practice, we find that strong language models with well instruction tuning adhere closely to this formatting instruction.

---

> ### Author Response · Authors · 2025-11-23
> **Rebuttal by Authors**
>
> **Q5**: Appendix F suggests TrigReason might lag behind LRM-only performance at very high token budgets (32K). Can you quantify this gap and explain why trigger-based collaboration might become suboptimal in resource-abundant settings?
>
> **R**: Indeed, as shown in Appendix F, when the token budget reaches 32K, the LRM-only baseline achieves an accuracy of 52.5%, while TrigReason and SpecReason yields 47.5%, resulting in a 5 percentage-point gap. We attribute this gap to two factors:
>
> (1) TrigReason’s core assumption is that many reasoning steps are routine and can be safely delegated to the SRM. However, under very high budgets, performance differences arise primarily on extremely complex problems that require long, intricate reasoning, where SRM assistance provides limited benefit.
>
> (2) With abundant tokens, the LRM can “brute-force” self-correction through exhaustive exploration and iterative refinement, making external collaboration suboptimal.
>
> TrigReason is designed for efficiency-constrained settings, aiming to improve efficiency while preserving performance. In resource-abundant scenarios where efficiency is not a concern, we fully agree that using the LRM alone is preferable, and we explicitly recommend this in practice.
>
> We hope this response has adequately addressed your concerns. If you have any further queries or require additional information, please feel free to let us know. We look forward to the opportunity for additional discussion during the review process.

---

> > ### Comment · Reviewer_EARS · 2025-11-25
> > **Thank you for the response.**
> >
> > Thank you to the authors for their detailed response and dedicated additional experiments. The response makes sense to me, and I do appreciate the idea of the paper. I lean towards acceptance of the paper and will maintain my score.

---

### Official Review · Reviewer_zBA1 · 2025-10-31

**Soundness:** 3
**Presentation:** 2
**Contribution:** 2
**Rating:** 4
**Confidence:** 5

**Summary:**

This paper introduces TrigReason, a trigger-based collaborative reasoning framework that coordinates Small Reasoning Models (SRMs) and Large Reasoning Models (LRMs). Unlike the polling-based paradigm used in SpecReason, which continuously queries the LRM for verification at each reasoning step, TrigReason employs event-driven triggers (strategic priming, cognitive offload, and intervention request) to selectively involve the LRM only when necessary. Empirical evaluations on AIME24, AIME25, and GPQA Diamond demonstrate that TrigReason reduces latency and API cost while maintaining comparable accuracy to full-LRM reasoning. The authors claim that this selective intervention mechanism effectively balances efficiency and reliability in collaborative reasoning.

**Strengths:**

1. Clear Problem Formulation – The paper provides a well-articulated motivation by analyzing inefficiencies in the existing polling-based speculative reasoning frameworks (e.g., SpecReason).
2. Systematic Characterization – The identification of three SRM failure modes (path divergence, cognitive overload, recovery inability) adds structure to the reasoning failure analysis.
3. Empirical Breadth – The experiments cover multiple datasets (AIME24/25, GPQA Diamond) and SRM–LRM pairs, showing generality in observed trends.

**Weaknesses:**

1. While the trigger-based scheme is presented as novel, its underlying principle (adaptive invocation of a stronger model based on simple heuristic signals) is an incremental modification over existing selective collaboration paradigms. The work lacks deeper theoretical or algorithmic contribution beyond heuristic event detection.
2. The triggers (confidence, hesitation words, initial planning) appear manually designed and dataset-specific. There is little discussion on how robust or general these heuristics are across reasoning domains (e.g., multi-modal reasoning, logic proofs, planning tasks).
3. Lack of Statistical Validation: The claimed efficiency improvements are presented mainly in relative percentages, without statistical significance testing or variance analysis. There is no examination of whether results are stable across random seeds or reasoning temperatures.
4. Incomplete Comparative Baseline: The experiments compare mainly with SpecReason and full LRM inference, but omit several recent acceleration or mixture frameworks (e.g., ReaLM, ThinkFlow, or self-adaptive CoT controllers). This makes it difficult to judge whether TrigReason truly advances the state of the art.
5. The paper excludes latency from the main results, citing hardware variability, yet latency is central to the efficiency motivation. This omission weakens the empirical foundation of the efficiency claims.
6. While three risk types are defined, their quantitative prevalence and interaction are not thoroughly analyzed. For example, it is unclear how often each trigger actually fires, and whether those interventions consistently lead to correct outcomes.

**Questions:**

1. Please report the frequency and distribution of each trigger’s activation (strategic priming, cognitive offload, intervention request). How do these correlate with actual accuracy gains or failures?
2. Have you tested whether the same thresholds (e.g., perplexity < 1.05, ρ = 0.85) transfer to unseen reasoning domains or models? Adding results on a dataset outside math reasoning (e.g., scientific QA, commonsense reasoning) would strengthen claims of generality.
3. Include comparisons with ReaLM (2025), ThinkFlow (2024), or Adaptive-CoT controllers, which also perform selective reasoning delegation. Without these, the novelty and empirical advantage are hard to assess.
4. Even if hardware-sensitive, please provide approximate latency distributions (mean, std) under identical conditions for fairness. Otherwise, efficiency claims remain qualitative.
5. What happens if hesitation-based triggers are replaced by learned detectors or simplified confidence thresholds? This would clarify whether the improvements stem from general mechanisms or specific handcrafted choices.

---

> ### Author Response · Authors · 2025-11-23
> **Rebuttal by Authors**
>
> Thank you for your detailed review and valuable feedback! Below, we address the comments you raised:
>
> **W1**: While the trigger-based scheme is presented as novel, its underlying principle (adaptive invocation of a stronger model based on simple heuristic signals) is an incremental modification over existing selective collaboration paradigms. The work lacks deeper theoretical or algorithmic contribution beyond heuristic event detection.
>
> **W4**: Incomplete Comparative Baseline: The experiments compare mainly with SpecReason and full LRM inference, but omit several recent acceleration or mixture frameworks (e.g., ReaLM, ThinkFlow, or self-adaptive CoT controllers). This makes it difficult to judge whether TrigReason truly advances the state of the art.
>
> **Q3**: Include comparisons with ReaLM (2025), ThinkFlow (2024), or Adaptive-CoT controllers, which also perform selective reasoning delegation. Without these, the novelty and empirical advantage are hard to assess.
>
> **R**: We appreciate the suggestion to compare against recent frameworks such as ReaLM (2025), ThinkFlow (2024), and adaptive CoT controllers. However, the cited works differ fundamentally from TrigReason: ReaLM [1] is a reinforcement learning (RL) framework designed to enhance reasoning of SRMs within vertical domains; it is not reasoning via SRM–LRM collaboration.  AdaCoT [2] is an RL-based training framework that enables LLMs to adaptively decide when to invoke CoT; it is training-based effiency method and does not involve collaboration between small and large models. As for ThinkFlow (2024), we are unable to locate the published paper matching this abbreviated name after extensive searching. These methods differ fundamentally from TrigReason in both problem formulation and technical approach.  TrigReason aims to improve reasoning efficiency by enabling collaboration between SRM and LRM while preserving output quality as much as possible. Built upon a key insight into the capability gap between SRM and LRM, TrigReason employs a training-free, trigger-based selective intervention mechanism to achieve the balance between effiency and accuracy. This collaborative paradigm is distinct from existing selective inference or adaptive CoT methods, which operate within a single model and often require training. If the works we identified do not match the reviewer’s intent, we would greatly appreciate clarification so we can correct our understanding.
>
> [1] ReaLM: Reflection-Enhanced Autonomous Reasoning with Small Language Models.
>
> [2] AdaCoT: Pareto-Optimal Adaptive Chain-of-Thought Triggering via Reinforcement Learning.
>
> **W2**: The triggers (confidence, hesitation words, initial planning) appear manually designed and dataset-specific. There is little discussion on how robust or general these heuristics are across reasoning domains (e.g., multi-modal reasoning, logic proofs, planning tasks).
>
> **Q2**: Have you tested whether the same thresholds (e.g., perplexity < 1.05, ρ = 0.85) transfer to unseen reasoning domains or models? Adding results on a dataset outside math reasoning (e.g., scientific QA, commonsense reasoning) would strengthen claims of generality.
>
> **R**: To evaluate the robustness of TrigReason across different domains, we further conduct experiments on two benchmarks beyond math reasoning:
> - **Big Bench Hard (BBH)**: a complex logical reasoning benchmark
> - **AI2 Reasoning Challenge (ARC-Challenge)**: a commonsense reasoning benchmark
>
> We use the same model pair as in the main results of TrigReason: DeepSeek-R1-1.5B (SRM) and QwQ-32B (LRM). Due to the lower task complexity compared to AIME, we adjusted the number of Priming Steps (n) from 20 to 5 to improve efficiency. All other parameters such as the Cognitive Overload Threshold (perplexity < 1.05, ρ = 0.85) and Rectification Steps (m = 1) are kept unchanged. The results are shown in the table below:
>
> | Metric     | SRM       | LRM       | TrigReason |
> |------------|-----------|-----------|------------|
> | Accuracy   |           |           |            |
> | BBH        | 42.2%     | 67.5%     | **68.7%**  |
> | ARC        | 59.3%     | 95.7%     | **94.8%**  |
> | Latency       |           |           |            |
> | BBH        | 34.4 min  | 178.2 min | **125.7 min** |
> | ARC        | 7.0 s     | 22.9 s    | **15.6 s**  |
>
> As shown, TrigReason achieves performance close to or even exceeding that of LRM reasoning on both tasks, while significantly outperforming SRM-only baselines, demonstrating its effectiveness in diverse domains.

---

> ### Author Response · Authors · 2025-11-23
> **Rebuttal by Authors**
>
> **W5**: The paper excludes latency from the main results, citing hardware variability, yet latency is central to the efficiency motivation. This omission weakens the empirical foundation of the efficiency claims.
>
> **Q4**: Even if hardware-sensitive, please provide approximate latency distributions (mean, std) under identical conditions for fairness. Otherwise, efficiency claims remain qualitative.
>
> **R**: We thank the reviewer for the feedback.  We report the latency for the evaluation on BBH and ARC benchmarks in the table in our response to W2/Q2 (above).
> - For **BBH**, we report the average latency per subset (across all 27 tasks).
> - For **ARC-Challenge**, we report the average latency per test instance.
>
> All experiments on BBH and ARC-Challenge are conducted on 8 NVIDIA RTX 4090 GPUs using SGLang v0.4.9 as the inference engine, with both SRM and LRM using tensor parallelism = 4. Otherwise, the mean latency comparison on AIME-24 dataset is shown in Figure 1(e) of our original paper. Together, they provide quantitative evidence of TrigReason’s efficiency gains across multiple benchmarks.
>
> **W3**: Lack of Statistical Validation: The claimed efficiency improvements are presented mainly in relative percentages, without statistical significance testing or variance analysis. There is no examination of whether results are stable across random seeds or reasoning temperatures.
>
> **R**: As stated in our Experiment Setup, we follow the standard evaluation protocol used in prior work [3,4]: we report pass@1 with k=16, which computes accuracy by sampling 16 independent reasoning trajectories per question. This approach inherently accounts for variability due to random seeds and decoding temperatures, as the final metric is an average over multiple stochastic runs, effectively smoothing out randomness and providing a stable, reproducible measure of performance.
>
> [3] Deepseek-r1: Incentivizing reasoning capability in llms via reinforcement learning.
>
> [4] Specreason: Fast and accurate inference-time compute via speculative reasoning.
>
> **W6**: While three risk types are defined, their quantitative prevalence and interaction are not thoroughly analyzed. For example, it is unclear how often each trigger actually fires, and whether those interventions consistently lead to correct outcomes.
>
> **Q1**: Please report the frequency and distribution of each trigger’s activation (strategic priming, cognitive offload, intervention request). How do these correlate with actual accuracy gains or failures?
>
> **R**: We analyze activation frequency of each trigger on the AIME24 dataset, as well as their correlation with final accuracy, and present the results in the table below.
>
> | Parameter (ρ-n-m) | Cognitive Offload (%) | Strategic Priming (%) | Intervention Request (%) | Total Trigger (%) | AIME24 ACC | LRM ACC |
> |-------------------|------------------------|------------------------|----------------------------|--------------------|------------|----------------|
> | 0.75-20-1         | 38.50%                 | 8.06%                  | 4.36%                      | 50.92%             | 30.0       |     29.2           |
> | 0.85-20-1         | 25.95%                 | 8.31%                  | 4.73%                      | 38.99%             | 29.6       |                |
> | 0.95-20-1         | 11.49%                 | 8.64%                  | 4.95%                      | 25.08%             | 26.25      |                |
> | 0.85-10-1         | 26.41%                 | 4.23%                  | 4.87%                      | 35.51%             | 27.5       |                |
> | 0.85-20-0         | 26.04%                 | 8.23%                  | 0.00%                      | 34.27%             | 23.33      |                |
>
> As shown:
> - The Cognitive Offload Trigger is the most frequently activated, accounting for over 25% of steps in most configurations.
> - The Intervention Request Trigger has a lower activation rate but contributes significantly to performance gains, enabling a performance jump from 23.33 to 29.6 accuracy when enabled (0.85-20-0 vs. 0.85-20-1), despite being triggered in only ~4.7% of steps.
> - Increasing trigger frequency generally improves accuracy, but with diminishing returns when it's close to LRM performance: for example, raising the cognitive offload threshold from 0.75 to 0.85 reduces its activation by \~12%, yet accuracy remains nearly unchanged.

---

> ### Author Response · Authors · 2025-11-23
> **Rebuttal by Authors**
>
> **Q5**: What happens if hesitation-based triggers are replaced by learned detectors or simplified confidence thresholds? This would clarify whether the improvements stem from general mechanisms or specific handcrafted choices.
>
> **R**: We thank the reviewer for this insightful suggestion. Hesitation-based triggers are designed based on efficiency considerations. Although they are effective in practice, they still raise concerns about generalization. Therefore, we design a hesitancy sentiment classifier to replace the detection of hesitation words. Specifically, we first collect 12k labeled reasoning steps from AIME, GPQA, and BBH, annotated using Qwen3-32B. And then, We fine-tune BERT-Tiny [5] (a lightweight model with only 2 transformer layers) for binary hesitation classification, using a linear classifier on top of the [CLS] token representation, following standard BERT-based sequence classification. Finally, We integrate this trained classifier into TrigReason’s pipeline to replace the original hesitation words based detector. The benchmark results are shown in the table below:
>
> | Parameter (ρ-n-m) | Cognitive Offload (%) | Strategic Priming (%) | Intervention Request (%) | Total Trigger (%) | AIME24 ACC | AIME25 ACC |
> |-------------------|------------------------|------------------------|----------------------------|--------------------|------------|------------|
> | 0.85-20-1 (original)       | 25.95%                 | 8.31%                  | 4.73%                      | 38.99%             | 29.6       | 29.58      |
> | 0.85-20-1 (with classifier) | 25.22%           | 8.47%                  | 6.21%                      | 39.90%             | 29.6       | 30.00  |
>
> It can be seen that the learned detector achieves higher activation frequency of Intervention Request, indicating broader coverage.
> It also leads to a slight improvement in accuracy on AIME25 dataset. We will make this learned trigger option configurable in TrigReason, allowing users to choose between heuristic and learned variants based on their needs.
>
> [5] Generalization in NLI: Ways (Not) To Go Beyond Simple Heuristics.
>
> We hope this response has adequately addressed your concerns. If you have any further queries or require additional information, please feel free to let us know. We look forward to the opportunity for additional discussion during the review process.

---

> > ### Comment · Reviewer_zBA1 · 2025-11-26
> >
> > Thank you for the detailed responses. Some concerns are partially resolved, especially regarding trigger activation statistics, cross-domain robustness, and the ablation with a learned hesitation detector. These additions would be valuable if incorporated into the main paper, rather than only in the rebuttal.
> >
> > However, several key issues remain insufficiently addressed:
> > * The novelty claim still lacks theoretical depth beyond heuristic trigger design. Clarifying the principled basis (e.g., capability gap modeling, intervention optimality, or error bounds) would strengthen the contribution.
> > * The baseline comparison remains incomplete. Even if ReaLM or AdaCoT differ in formulation, reporting results—or a reimplementation—would better contextualize TrigReason’s advantage.
> > * The latency evidence is still limited. Distribution (mean ± std) under identical configuration is necessary to support efficiency claims.
> > * Statistical validation (variance, multiple seeds, temperature sensitivity) is not covered by reporting pass@1 with sampling.
> >
> > If these aspects are expanded in the revised manuscript, the paper will be considerably stronger.

---

### Note · Authors · 2026-01-05

I have read and agree with the venue's withdrawal policy on behalf of myself and my co-authors.